# Bestrophin-4 relays HES4 and interacts with TWIST1 to suppress epithelial-to-mesenchymal transition in colorectal cancer cells

Zijing Wang[1], Bihan Xia[1], Shaochong Qi[1], Xian Zhang[1], Xiaoshuang Zhang[1], Yan Li[1], Huimin Wang[1], Miao Zhang[2,3], Ziyi Zhao[4], David Kerr[2], Li Yang[1], Shijie Cai[2]*, Jilin Yang[1]*

[1]Department of Gastroenterology and Hepatology, Sichuan University-University of Oxford Huaxi Joint Centre for Gastrointestinal Cancer, West China Hospital, Sichuan University, Chengdu, China; [2]Nuffield Division of Clinical Laboratory Sciences, Radcliffe Department of Medicine, University of Oxford, Oxford, United Kingdom; [3]College of Acupuncture and Moxibustion, Fujian University of Traditional Chinese Medicine, Fuzhou, China; [4]TCM Regulating Metabolic Diseases Key Laboratory of Sichuan Province, Hospital of Chengdu University of Traditional Chinese Medicine, Chengdu, China

*For correspondence:
shijie.cai@ndcls.ox.ac.uk (SC);
yangjinlin@wchscu.cn (JY)

Competing interest: The authors declare that no competing interests exist.

## eLife Assessment

The findings of this **valuable** manuscript advance our understanding of the significance of Bestrophin isoform 4 (BEST4) in suppressing colorectal cancer (CRC) progression. The authors used appropriate and validated methodology, such as the knockout of BEST4 using CRISPR/Cas9 in CRC cells, to provide a **solid** foundation for elucidating the potential link between BEST4 and CRC progression.

**Abstract** Bestrophin isoform 4 (*BEST4*) is a newly identified subtype of the calcium-activated chloride channel family. Analysis of colonic epithelial cell diversity by single-cell RNA-sequencing has revealed the existence of a cluster of *BEST4*+ mature colonocytes in humans. However, if the role of *BEST4* is involved in regulating tumour progression remains largely unknown. In this study, we demonstrate that *BEST4* overexpression attenuates cell proliferation, colony formation, and mobility in colorectal cancer (CRC) in vitro, and impedes the tumour growth and the liver metastasis in vivo. BEST4 is co-expressed with hairy/enhancer of split 4 (*HES4*) in the nucleus of cells, and HES4 signals *BEST4* by interacting with the upstream region of the *BEST4* promoter. *BEST4* is epistatic to *HES4* and downregulates TWIST1, thereby inhibiting epithelial-to-mesenchymal transition (EMT) in CRC. Conversely, knockout of BEST4 using CRISPR/Cas9 in CRC cells revitalises tumour growth and induces EMT. Furthermore, the low level of the *BEST4* mRNA is correlated with advanced and the worse prognosis, suggesting its potential role involving CRC progression.

## Introduction

Colorectal cancer (CRC) is a common tumour worldwide. A steady increase in the morbidity and mortality of CRC has been reported (*Rawla et al., 2019*). When diagnosed in advanced stages,

epithelial-mesenchymal transition (EMT) may occur as CRC metastasize to distal organs (*Pastushenko and Blanpain, 2019*; *Sunlin Yong et al., 2021*; *Yeung and Yang, 2017*; *Zhang et al., 2021*). EMT is a reprogramming process in which epithelial E-cadherin and TJP1 are suppressed, and mesenchymal N-cadherin and VIM are induced, transcriptionally mediated by Twist and the Snail family (*Dongre and Weinberg, 2019*). Overexpression of TWIST1 significantly enhances the migration and invasion capabilities of CRC cells; furthermore, it is closely associated with metastasis and poor prognosis in patients with CRC (*Yusup et al., 2017*; *Zhu et al., 2015*).

The inhibition of EMT in breast cancer and prostate cancer has been associated with calcium-activated chloride channels (CaCCs) (*Porretti et al., 2018*; *Walia et al., 2012*), leading to various evaluations of CaCCs as a potential for cancer therapy (*Li et al., 2017*; *Miotto and Struhl, 2006*; *Qu et al., 2019*; *Sui et al., 2014*). The bestrophin (*BEST*) chloride channel, which is a member of a recently identified gene family, encodes several integral membrane proteins expressed on the baso-lateral membranes of epithelial cells; they regulate CaCCs to maintain homeostasis (*Marmorstein et al., 2009*; *Tsunenari et al., 2003*). Although there are four human evolutionarily related genes (*BEST1*–*BEST4*) (*Miller et al., 2019*; *Tsunenari et al., 2003*), *BEST4* is expressed predominantly in the colon (*Fagerberg et al., 2014*), where it gates electrolyte transportation in human intestinal epithelial cells (*Ito et al., 2013*). However, its role and the mechanism in regulating the development of CRC are currently unknown.

In a recent analysis based on single-cell RNA-sequencing analysis, it was discovered that a group of mature colonocytes expresses *BEST4* (*Parikh et al., 2019*), and the subset of this group coexists with hairy/enhancer of split 4 positive (*HES4*⁺) cells (*Parikh et al., 2019*). *HES4* belongs to a family of seven Hes genes (Hes1–7) that encodes a transcription factor with a basic helix-loop-helix motif. It is known to play a critical role in determining the fate and differentiation of progenitor cells during the development of various organs. The previous study has demonstrated the antagonistic role of HES4 in regulating TWIST1 through protein-protein interaction, which governs the differentiation of bone marrow stromal/stem cell lineage (*Cakouros et al., 2015*). Inhibiting the TWIST1 transcriptional factor has been shown to mediate commitment to the differentiation of human bone marrow stem cells, as reported by several studies (*Cakouros et al., 2015*; *El Yakoubi et al., 2012*; *Murato and Hashimoto, 2009*; *Nichane et al., 2008*). Given this information, it is tempting to speculate that BEST4 may be associated with HES4 and TWIST1 in regulating EMT during CRC progression.

The objectives of the present study are as follows: (1) to establish the role of BEST4 in CRC growth both in vitro and in vivo; (2) to determine the underlying molecular mechanisms by which BEST4 inter-acts with HES4 and TWIST1, thereby inducing EMT; and (3) to investigate the correlation between BEST4 expression and prognosis of CRC.

## Results

### *BEST4* inhibits CRC proliferation, clonogenesis, migration, and invasion in vitro

To evaluate the effect of BEST4 on tumour growth, we constructed *BEST4*-expressing HCT116 and Caco2 CRC cell lines (*Figure 1—figure supplement 1*) and showed that *BEST4* overexpression halved cell proliferating rates of their respective empty vector (EV) controls when they were monitored in live and analysed using InCucyte (*Figure 1A and B*). This was accompanied by decreased cell viability (*Figure 1—figure supplement 1*). To determine the causal role of BEST4 in CRC tumourigenesis, we knocked out the *BEST4* in HCT-15 using a CRISPR/Cas9 system (*Figure 1—figure supplement 1*). Consequently, there was a 50% increase in proliferation rates and viability compared with the parental control (*Figure 1C and D*; *Figure 1—figure supplement 1*). Rescued expression of BEST4 was able to reverse the observed effects on HCT-15 (*Figure 1D*; *Figure 1—figure supplement 1*).

In contrast to the controls, BEST4 overexpression decreased colony formations of HCT116 and Caco2 by 60% and 70%, respectively (*Figure 1E*). While BEST4 deletion increased the number of colonies by fivefold in HCT-15 (*Figure 1F*), the rescued expression counteracted its enhancement (*Figure 1F*).

Furthermore, overexpression of *BEST4* in HCT116 inhibited transwell migration and invasion by 70 and 80%, respectively, compared to EV controls (*Figure 1G*). But *BEST4* ablation in HCT-15

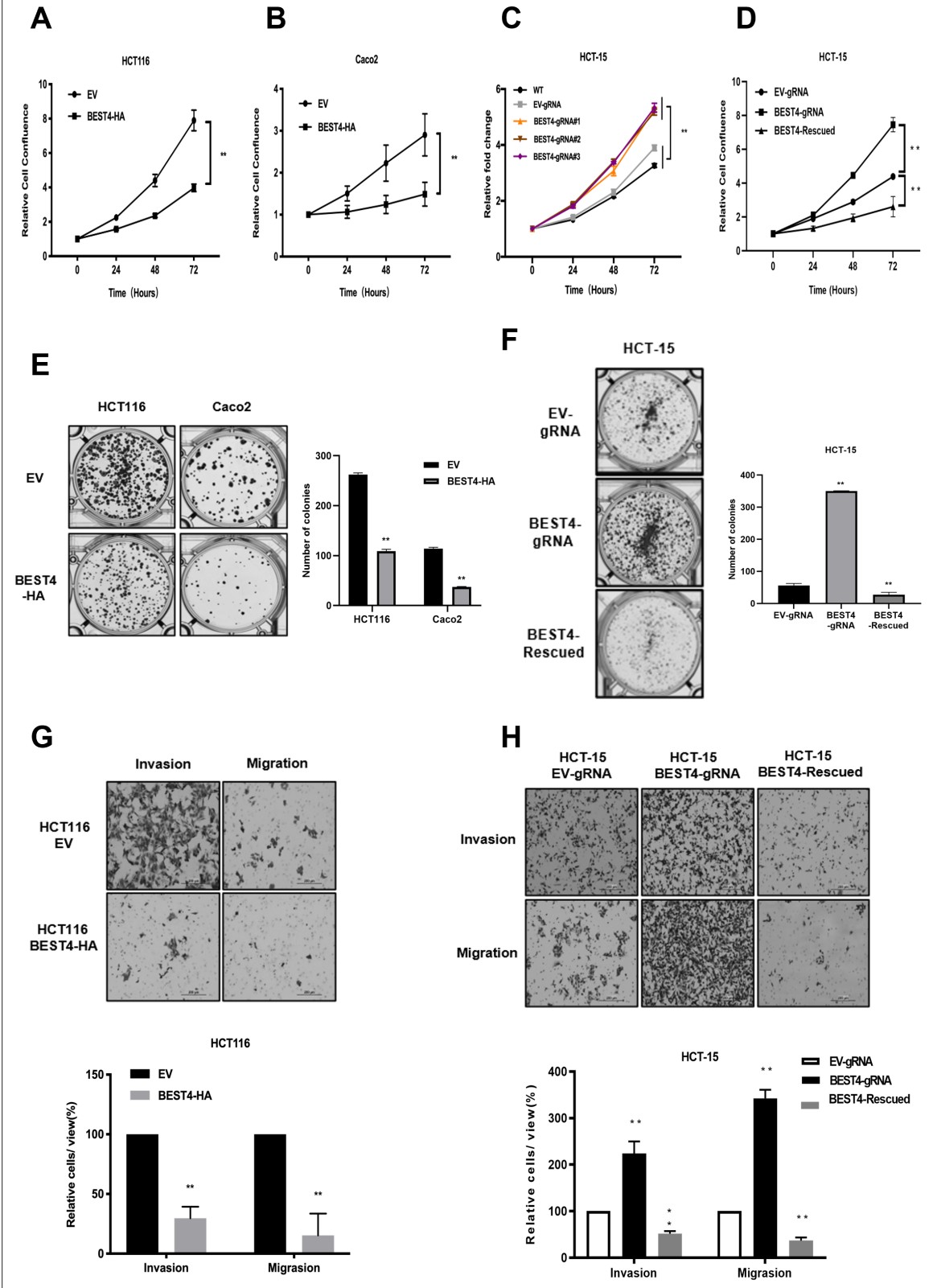

**Figure 1.** *BEST4* inhibits colorectal cancer (CRC) cell proliferation, clonogenesis, migration, and invasion in vitro. Overexpression of *BEST4* decreased proliferation of HCT 116 (**A**) and Caco2 (**B**) cells as determined by the IncuCyte confluence assay. (**C**) Viability of three individual *BEST4*-deleted clones as determined by a CCK-8 assay. (**D**) *BEST4* deletion significantly enhanced cell viability compared with cells transduced with empty vector (EV)-gRNA control; the reintroduction of *BEST4* resulted in suppressed cell growth as determined by IncuCyte confluence analysis. (**E**) Expression of *BEST4*-HA in

*Figure 1 continued on next page*

*Figure 1 continued*

HCT116 and Caco2 cells resulted in more colonies than cells transduced with EV plasmids (left panel); colonies were counted and analysed as shown (right panel). (**F**) A colony formation assay was performed to evaluate the responses of HCT-15 cells following *BEST4* deletion and reintroduction (left panel); cell colonies were counted and analysed as shown (right panel). (**G**) Heterologous expression of BEST4 in HCT116 cells suppressed invasion and migration (upper panel, scale bar, 200 µm); cells that passed through the membrane were counted and compared (lower panel). (**H**) *BEST4* deletion in HCT-15 cells promoted invasion, whereas reintroduction of *BEST4* expression resulted in the inhibition of HCT-15 cell invasion and migration (upper panel, scale bar, 200 µm). Cells that passed through the membrane were counted and compared (lower panel). Data shown are the result of at least three independent experiments, with the mean ± SEM; *p<0.05, **p<0.01 vs EV or wild-type (WT).

The online version of this article includes the following source data and figure supplement(s) for figure 1:

**Figure supplement 1.** Construction of *BEST4*-overexpressing /*BEST4*-depetion colorectal cancer (CRC) cell lines, and cell proliferation measured by CCK-8 assay.

**Figure supplement 1—source data 1.** Original files for western blot analysis displayed in *Figure 1—figure supplement 1*.

**Figure supplement 1—source data 2.** PDF file containing original western blots for *Figure 1—figure supplement 1*, indicating the relevant bands and treatments.

increased cell migration and invasion by twofold and threefold, respectively. These pro-tumour effects on HCT-15 were also reversible when BEST4 expression was rescued (*Figure 1H*).

Collectively, these findings suggest the functional inhibitory effects of BEST4 on CRC development in vitro.

## *BEST4* deters EMT in CRC in vitro and in vivo

Because EMT initiation is a critical process in tumour progression (*Acloque et al., 2009*), we evaluated the effect of *BEST4* on EMT using quantitative polymerase chain reaction (qPCR) and found that *BEST4* expression upregulated *CDH1* and *TJP1* and downregulated *VIM* and *TWIST1* in HCT116 compared with the control (p<0.05) (*Figure 2A*). Conversely, knockout of *BEST4* in HCT15 downregulated *CDH1* and *TJP1* and upregulated *VIM* and *TWIST1*. Rescued expression of BEST4 in HCT-15 reversed these gene alterations (*Figure 2B*). Changes of these genes were consistent with their corresponding protein expression as analysed by immunoblotting (*Figure 2C and D*; *Figure 2—figure supplement 1C*).

Next, we subcutaneously implanted BEST4-HA-expressing HCT116 and EV control cell lines into BALB/c nude mice and monitored tumour growth weekly. When they reached the maximum allowable volumes (day 32), the animals were culled, and tumours were harvested. Essentially, the sizes and weights of tumours in *BEST4* overexpression group were only half of the EV control (*Figure 2E and F*), in conjunction with the similar magnitude reduction of Ki67+ cells as analysed by immunohistochemistry (*Figure 2G and H*). Further analysis of the HCT116 tumour tissue lysates showed that BEST4 deterred EMT by upregulating E-cadherin, and downregulating VIM and TWIST1 (*Figure 2I*; *Figure 2—figure supplement 1D*).

We also evaluated the inhibitory role of BEST4 in a model of CRC to the liver metastasis by intrasplenically injecting HCT-15 cell lines with BEST4 knockout (BEST4gRNA) or the wild-type control (gRNA) or the knockout with the rescue (BEST4-Rescued) into BALB/c nude mice. We observed that there were twofold greater of the liver metastatic nodules in the BEST4 deficient tumours than the control (*Figure 2J–L*). Conversely, the reconstituted BEST4 expression had no visible metastatic nodules in the liver, and the pathological analysis of the tissue sections by H&E staining also confirmed no detectable metastatic loci in the tissue sections (*Figure 2J–L*).

These findings indicate the role of BEST4 in suppressing CRC growth, through counteracting EMT induction in models of in vitro and in vivo.

## HES4 upstream regulation of BEST4

Given the co-expression of *BEST4*+ and *HES4*+ in a human colonic epithelial lineage (*Parikh et al., 2019*), we sought to investigate their biological interactions during tumour growth. The human CRC cell line, HCT116, was chosen for HES4 overexpression due to its relatively lower expression of HES4 compared to other CRC cell lines. Similarly, the LS174T cell line derived from human CRC was selected for *HES4* knockdown because it exhibited comparatively higher expression of *HES4* in comparison to other CRC cell lines (*Figure 3—figure supplement 1A*). Upon transfecting the Flag-*HES4* plasmid DNA into HCT116, we observed a responsive upregulation of both endogenous *BEST4* mRNA and

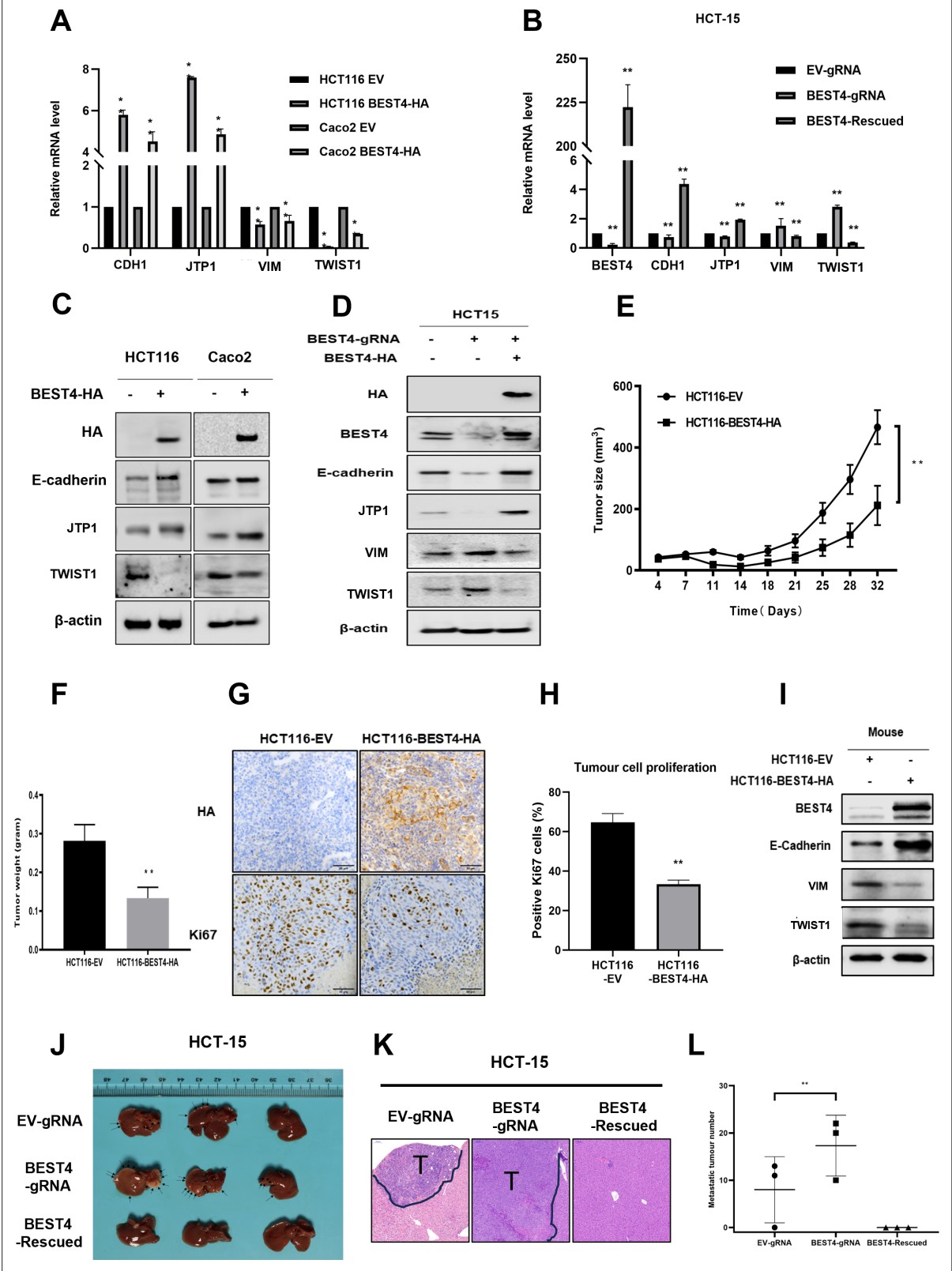

**Figure 2.** *BEST4* deters epithelial-to-mesenchymal transition (EMT) in colorectal cancer (CRC) in vitro and in vivo. (**A**) Expression of mRNAs encoding EMT-related genes as determined by quantitative polymerase chain reaction (qPCR) in HCT116 cells transfected with BEST4-HA or empty vector (EV). (**B**) As determined by qPCR, the expression of mRNAs encoding EMT markers in *BEST4*-deleted and *BEST4*-restored HCT-15 cells. (**C**) Protein expression of EMT-related genes was evaluated by western blotting of lysates of HCT116 cells transfected with BEST4-HA or EV. (**D**) Western blotting determined

*Figure 2 continued on next page*

*Figure 2 continued*

the levels of immunoreactive EMT markers in *BEST4*-deleted and *BEST4*-restored HCT-15 cells. (**E**) Growth of xenograft tumours in BALB/c nude mice resulting from the injection of HCT116 cells stably transduced with *BEST4*-HA was significantly diminished compared with those resulting from HCT116 cells transduced with EV alone (eight mice per group). (**F**) The histogram represents the mean weights of tumours isolated from the HCT116-*BEST4*-HA and HCT116-EV groups. (**G**) Tumours from the HCT116-*BEST4*-HA and HCT116-EV groups were sectioned and subjected to immunohistochemical staining to detect HA and human Ki67; scale bar = 50 μm. (**H**) Mean percentage of Ki67-positive staining in each group. (**I**) Representative western blot documenting protein levels of EMT markers in individual HCT116-BEST4 and HCT116-EV xenograft tumours. *p<0.05, **p<0.01. Data from at least three independent experiments are presented as the mean ± SEM. (**J–L**) Intrasplenical injections of HCT-15 cell lines with EV-gRNA control, BEST4-gRNA, or BEST4-Rescued (n=3 per group). After 28 days, the animals were sacrificed and numbers of metastatic nodules in the liver were counted, and the liver tissues were fixed for sectioning and H&E staining; T, tumour. Data are presented as the mean ± SEM. **p<0.01 vs EV-gRNA.

The online version of this article includes the following source data and figure supplement(s) for figure 2:

**Source data 1.** Original files for western blot analysis displayed in *Figure 2C, D, and I*.

**Source data 2.** PDF file containing original western blots for *Figure 2C, D, and I*, indicating the relevant bands and treatments.

**Figure supplement 1.** Western blot signals were quantified using ImageJ software.

protein levels compared to the control transfectant (*Figure 3A*; *Figure 3—figure supplement 1B and C*). This intrinsic association was further supported by knocking down the endogenous HES4 with two pairs of short hairpin RNAs (shRNAs) in LST174 that resulted in downregulation of endogenous BEST4 (*Figure 3B*; *Figure 3—figure supplement 1D and E*).

To explore how HES4 regulates *BEST4* expression at the transcriptional level, we transfected a *BEST4* promoter-driven luciferase DNA construct into *HES4*-expressing HCT116 and assessed the promoter activity using a dual-luciferase assay. Our results showed a fourfold increase in the promoter activity compared to the EV control (*Figure 3C*), Furthermore, chromatin immunoprecipitation-quantitative PCR (ChIP-qPCR) analysis using a Flag antibody to IP for HES4 revealed a 12-fold enrichment of the P3 region upstream of the *BEST4* transcriptional start site compared to the control (*Figure 3D*). Interestingly, *BEST4* overexpression in HCT116 cells did not affect *HES4* mRNA expression as quantified by qPCR (*Figure 3—figure supplement 1F*), suggesting that *HES4* regulates *BEST4* expression upstream. This speculation is supported by immunofluorescence staining, where we observed nuclear colocalisation of BEST4 (green) and HES4 (red) in HCT116 cells co-transfected with *BEST4*-HA and Flag-*HES4* plasmid DNAs, along with additional cytoplasmic expression of *BEST4* (*Figure 3E*).

To evaluate the association of *BEST4* with *HES4* at the translational level, we co-transfected their HA-tagged *BEST4* and Flag-*HES4* plasmid DNAs into HEK293 cells, along with the respective single transfectant control. By conducting IP with antibodies specific to Flag or HA, we showed that BEST4 and HES4 were reciprocally copurified from the nuclear protein lysates, but not with IgG controls (*Figure 3F*).

Collectively, these findings indicate that BEST4 interacts with HES4 at both transcriptional and translational levels.

## BEST4 relays HES4 signal and downregulates TWIST1 leading to EMT inhibition independent of the channel function in CRC

Because a step of HES4 in committing to human bone marrow stromal/stem cell lineage-specific development is mediated by TWIST1 downregulation (*Cakouros et al., 2015*), with evidence of direct interaction between BEST4 and HES4 observed in HCT116, it is plausible that they could exploit TWIST1 to regulate EMT. To test this notion, we transfected a *TWIST1* promoter-driven luciferase DNA construct into HCT116. In comparison with the EV control, Flag-*HES4* or *BEST4*-HA transfectants significantly decreased *TWIST1* promoter activity by almost half, as evaluated by the dual-luciferase assay (*Figure 4A*). Essentially, the *BEST4* transfectant enhanced inhibition of the *TWIST1* promoter by 70% in *HES4*-expressing HCT116 (*Figure 4B*; *Figure 4—figure supplement 1A*), hence suggesting that *BEST4* mediates the inhibition of the *TWIST1* promoter activity by *HES4*. This effect was also observed when endogenous *HES4* was knocked down using shRNAs in HCT116 cells transfected with *BEST4*-HA plasmid DNA (*Figure 4C*).

The association of *BEST4* with the *TWIST1* promoter activity was also evidenced by their nuclear colocalisation after transfecting *BEST4*-HA and Myc-*TWIST1* plasmid DNAs into HCT116 (*Figure 4—figure supplement 1B*). Co-immunoprecipitation (Co-IP) with antibodies to Myc or HA showed that they were copurified from the nuclear protein lysates, but not with the IgG controls from the

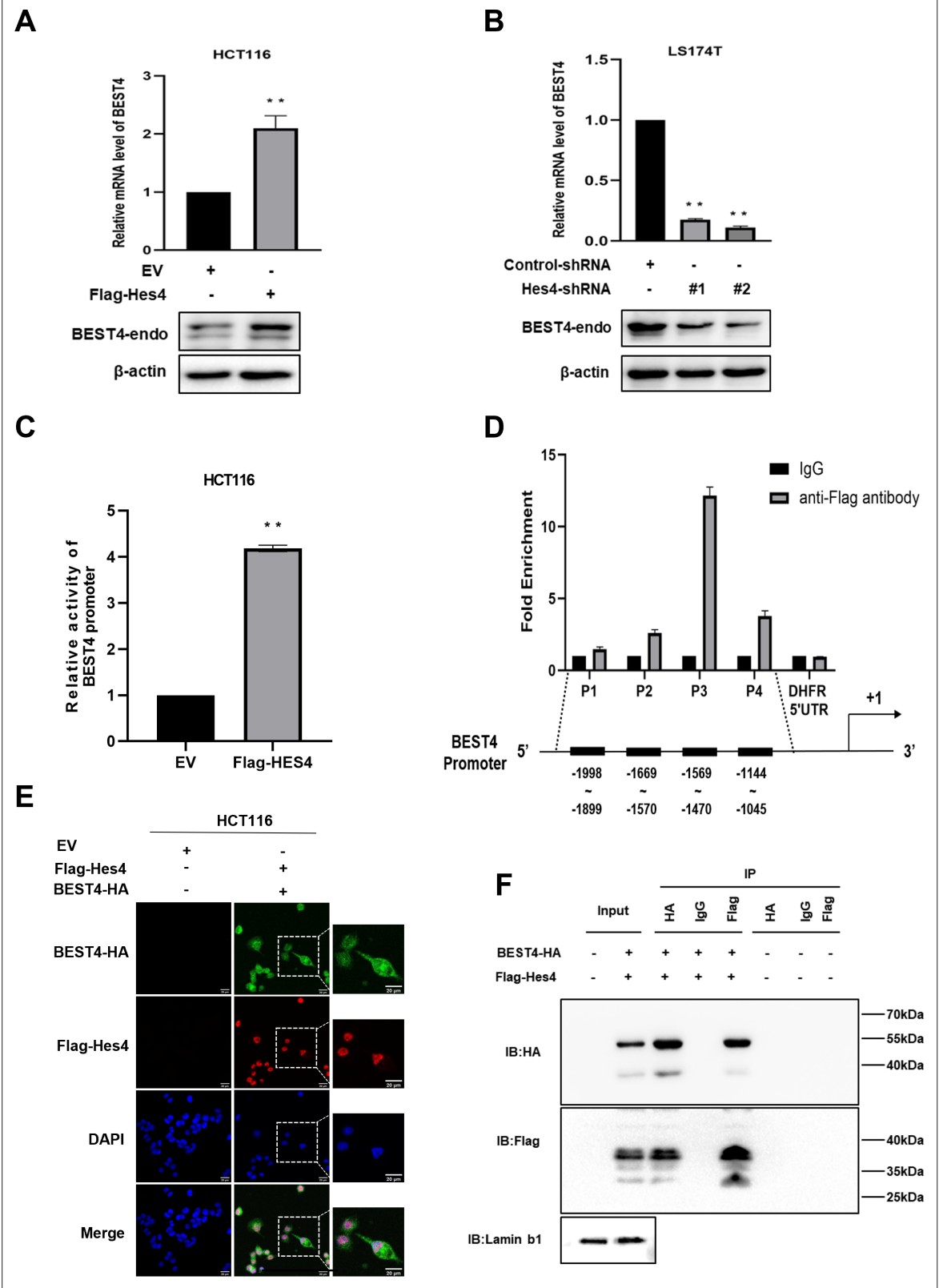

**Figure 3.** Molecular regulation of BEST4 expression by HES4. (**A**) Endogenous *BEST4* mRNA and protein levels in *HES4*-overexpressing HCT116 cells were detected by quantitative polymerase chain reaction (qPCR) and western blotting, respectively. (**B**) Endogenous *BEST4* mRNA and protein levels in LS174T cells after short hairpin RNA (shRNA)-mediated HES4 knockdown as detected by qPCR and western blotting, respectively. (**C**) Detection of *BEST4* promoter activity in HES4-overexpressing HCT116 cells using a dual-luciferase reporter assay. (**D**) As determined by chromatin

*Figure 3 continued on next page*

*Figure 3 continued*

immunoprecipitation (ChIP)-qPCR, fold-enrichment of the *BEST4* promoter region in Flag-ChIP samples from HES4-overexpressing HCT116 cells. Data were normalised using a fold-enrichment method (i.e. ChIP signals divided by control IgG signals). The DHFR 5' UTR was used as a negative control (upper panel). The primer sets that target the BEST4 promoter in the ChIP-qPCR assays are as illustrated; +1 marks the transcriptional start site of *BEST4*, and P1, P2, P3, and P4 represent primer locations (lower panels). (**E**) Immunofluorescence staining documents the colocalisation of BEST4 (green) and HES4 (red) in HCT116 cells based on detection of their specific tags; scale bar, 20 μM. (**F**) Co-immunoprecipitation of nuclear extracts with specifically tagged BEST4 and HES4 as determined by immunoblotting. Lamin B1 served as a nuclear protein control. (**G**) Detection of *BEST4* promoter activity in *HES4*-overexpressing HCT116 cells after *BEST4* knockdown through a dual-luciferase reporter assay. Data shown are the result of at least three independent experiments, with the mean ± SEM; *p<0.05, **p<0.01 vs empty vector (EV).

The online version of this article includes the following source data and figure supplement(s) for figure 3:

**Figure supplement 1—source data 1.** Original files for western blot analysis displayed in *Figure 3—figure supplement 1C and E*.

**Figure supplement 1—source data 2.** PDF file containing original western blots for *Figure 3—figure supplement 1C and E*, indicating the relevant bands and treatments.

**Source data 1.** Original files for western blot analysis displayed in *Figure 3A, B, and F*.

**Source data 2.** PDF file containing original western blots for *Figure 3A, B, and F*, indicating the relevant bands and treatments.

**Figure supplement 1.** *BEST4* interacted with *HES4*.

transfectant cell lysates (*Figure 4D*; *Figure 4—figure supplement 1B*). Additionally, the knockdown of endogenous *BEST4* in HES4-expressing HCT116 cells substantially decreased Flag-HES4 coprecipitation from the nuclear protein lysates, while Myc-TWIST1 expression remained constant, as determined by co-IP with antibodies to Flag or My (*Figure 4E*; *Figure 4—figure supplement 1C*).

We then investigated the impact of HES4 expression on EMT induction by overexpressing HES4 in HCT116 cells. Our results showed there were upregulation of TJP1 and E-cadherin epithelial molecules, and downregulation of TWIST1 protein (*Figure 4F*; *Figure 4—figure supplement 2A*) and the *VIM* mRNA (*Figure 4—figure supplement 1D*). The VIM protein appeared to be undetectable in HCT116, which was consistent with a previous report (*Roger et al., 2010*). Nevertheless, knocking down endogenous *HES4* in LS174T cells using shRNAs targeting two *HES4* domains caused the opposite effect (*Figure 4G*; *Figure 4—figure supplement 2B*).

To determine if BEST4 is downstream signal that relays HES4 to regulate TWIST1 expression and consequently EMT induction, we knocked down endogenous *BEST4* with RNAs silencing (siRNAs) in HES4-expressing HCT116 and showed the corresponding downregulation of E-cadherin and TJP1, with upregulation of TWIST1 protein and the *VIM* mRNA, as analysed by immunoblotting and qPCR (*Figure 4H*; *Figure 4—figure supplement 2C*; *Figure 4—figure supplement 1E and D*). The opposite was true when BEST4 was overexpressed in LS174T (*Figure 4I*; *Figure 4—figure supplement 2D*). Hence, inhibition of EMT induction by HES4 is BEST4-mediated by downregulation of TWIST1 expression in CRC.

Given the physiological role of *BEST4* plays in regulating chloride/calcium flux, we evaluated the channel functional effect on EMT induction. After respectively incubating BEST4-expresing HCT116 and the EV control with 300 μM DIDS (a broad blocker of chloride channel), or a gradient increase in doses of CaCCinh-A01 (a specific inhibitor of native CaCCs), or 1 μM ionomycin (a calcium ionophore to activate the chloride channel) as published previously (*Liu et al., 2015*), we showed that *BEST4* significantly increased chloride/calcium fluxes and activated their exchanges and the expected inhibition by DIDS, CaCCinh-A01, or ionomycin (*Figure 5—figure supplement 1C*). Through immunoblotting analysis, we recapitulated that BEST4 simultaneously upregulated both TJP1 and E-cadherin, and downregulated TWIST1. Essentially, the levels of these proteins remained constantly expressed when the cells were treated with DIDS or CaCCinh-A01 or ionomycin, respectively (*Figure 5—figure supplement 1F*). Hence, these results suggest *BEST4* protein-mediated EMT suppression is independent of its channel function in HCT116.

## Correlation of the *BEST4* with clinicopathology and outcomes of patients diagnosed with CRC

We showed evidence of the endogenous *BEST4* mRNA and protein expression in HCT116 and LS174T CRC cell lines (*Figure 3A and B*); these findings prompted us to explore the potential clinically significant *BEST4* expression in CRC. Initially with high-throughput whole transcriptome sequencing analysis

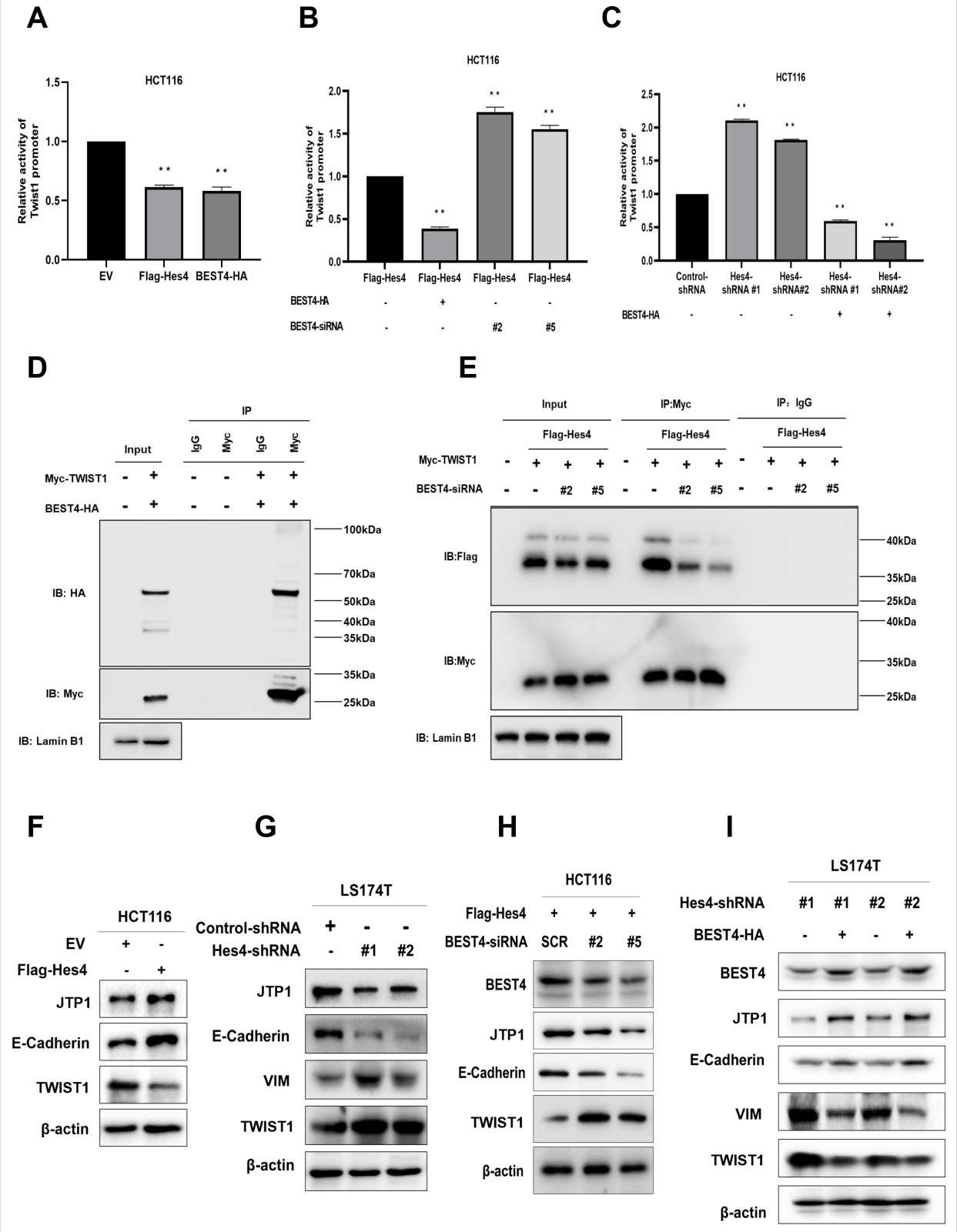

**Figure 4.** *BEST4* relays the HES4 signal and downregulates TWIST1 expression to counteract epithelial-to-mesenchymal transition (EMT) induction in colorectal cancer (CRC). (**A**) Detection of TWIST1 promoter activity in *HES4*-overexpressing or *BEST4*-overexpressing HCT116 cells through a dual-luciferase reporter assay. (**B**) Detection of TWIST1 promoter activity in *HES4*-overexpressing HCT116 cells after BEST4 knockdown through a dual-luciferase reporter assay. (**C**) Detection of TWIST1 promoter activity in HCT116 cells after short hairpin RNA (shRNA)-mediated HES4 knockdown and

*Figure 4 continued on next page*

*Figure 4 continued*

combination with *BEST4*-overexpressing using a dual-luciferase reporter assay. (**D**) Co-immunoprecipitation of nuclear extracts with specifically tagged BEST4 and TWIST1 as determined by immunoblotting. Lamin B1 served as a nuclear protein control. (**E**) Co-immunoprecipitation of nuclear extracts with specifically tagged HES4 and TWIST1 as determined by immunoblotting. Lamin B1 served as a nuclear protein control. (**F**) Expression of EMT markers in *HES4*-overexpressing HCT116 cells. (**G**) Expression of EMT markers in LS174T cells after shRNA-mediated *HES4* knockdown. (**H**) Expression of EMT markers in *HES4*-overexpressing HCT116 cells after BEST4 knockdown. (**I**) Expression levels of EMT markers in *BEST4*-overexpressing LS174T cells after HES4 knockdown; SCR, scrambled siRNA.

The online version of this article includes the following source data and figure supplement(s) for figure 4:

**Source data 1.** Original files for western blot analysis displayed in *Figure 4D–I*.

**Source data 2.** PDF file containing original western blots for *Figure 4D–I*, indicating the relevant bands and treatments.

**Figure supplement 1.** *BEST4* interacted with *HES4* to inhibit TWIST1.

**Figure supplement 1—source data 1.** Original files for western blot analysis displayed in *Figure 4—figure supplement 1B and C*.

**Figure supplement 1—source data 2.** PDF file containing original western blots for *Figure 4—figure supplement 1B and C*, indicating the relevant bands and treatments.

**Figure supplement 2.** Western blot signals were quantified using ImageJ software.

of five pairs of CRCs and adjacent normal tissues (ANTs), we identified several significantly down-regulated genes in CRC compared to ANTs; *BEST4* was in the top four (*Figure 5A and B*). Further validation by qPCR in the paired tissues of 50 colorectal adenomas and 124 CRC showed that the *BEST4* was expressed significantly less in adenomas and tumours than in ANTs (*Figure 5C and D*). Immunoblotting analysis of tissue lysates from five paired randomly selected samples showed that BEST4 protein was downregulated in CRC (*Figure 5E*; *Figure 5—figure supplement 1G*). Using public datasets of The Cancer Genome Atlas (TCGA) and the Genotype-Tissue Expression (GTEx), we measured *BEST4* mRNA expression in CRC and matched normal tissues with the online Gene Expression Profiling Interactive Analysis (GEPIA) tool (http://gepia.cancer-pku.cn/). As shown in *Figure 6A* the normal colorectal control tissues consistently expressed higher *BEST4* than the CRC.

We further evaluated the clinicopathological significance of *BEST4* expression in CRC patients by examining its correlation with gender, tumour-node-metastases stage, and lymph node metastasis based on the eighth edition of the American Joint Committee on Cancer Staging Manual (*Amin, 2017*). Interestingly, low *BEST4* mRNA expression in males was twice as high as in females (p=0.022, *Table 1*). Accordingly, low *BEST4* mRNA expression correlated with more advanced stages (p=0.02, *Table 1*) and lymph node metastasis (between 0 and 2) (p=0.032, *Table 1*). Patients with high *BEST4* expression had better overall survival than those with low expression (p<0.01, *Figure 5F*), which was confirmed by univariate survival analysis showing a lower risk of death (p<0.001, *Table 2*). Multivariate Cox regression analysis indicated that low levels of BEST4 mRNA were associated with poor prognosis of CRC patients (hazard ratio [HR], 0.285; 95% confidence interval, 0.157–0.606; p<0.001; *Table 2*).

After the significant reduction of the *HES4* mRNA expression through RNA-sequencing analysis of our CRC patients (*Figure 6B and C*), we proceeded to investigate the connection between *BEST4* and *HES4* in TCGA and GTEx (http://gepia.cancer-pku.cn/). Our findings revealed a positive correlation between these genes (R=0.47, p<0.001) (*Figure 6D*), providing further evidence of their association in CRC. Furthermore, we conducted additional mining of TCGA or GTEx datasets using the *GEPIA tool* (http://gepia.cancer-pku.cn/) and observed a positive correlation between *BEST4* and the *CDH1* mRNA (R=0.77, p<0.001), with a negative correlation with *VIM* (R = –0.49, p<0.001) and *TWIST1* mRNA (R = –0.39, p<0.001) (*Figure 6E–G*). These correlation analyses of RNA-sequencing data from our patients and publicly available datasets underscore the association of low *BEST4* expression with worse outcomes in malignant CRC.

## Discussion

In cancer cell cultures and mouse tumour xenografts, the present study shows that *BEST4* inhibits the development of CRC. An axis comprising *HES4/BEST4/ TWIST1* governs the induction of EMT in CRC. *BEST4* interacts with *HES4* at the transcriptional and translational levels, and downregulates TWIST1 expression, resulting in EMT inhibition but independent of its channel functions. Our findings support the notion in several ways: (1) *HES4* binds to the promoter region of *BEST4*; (2) BEST4 colocalises with

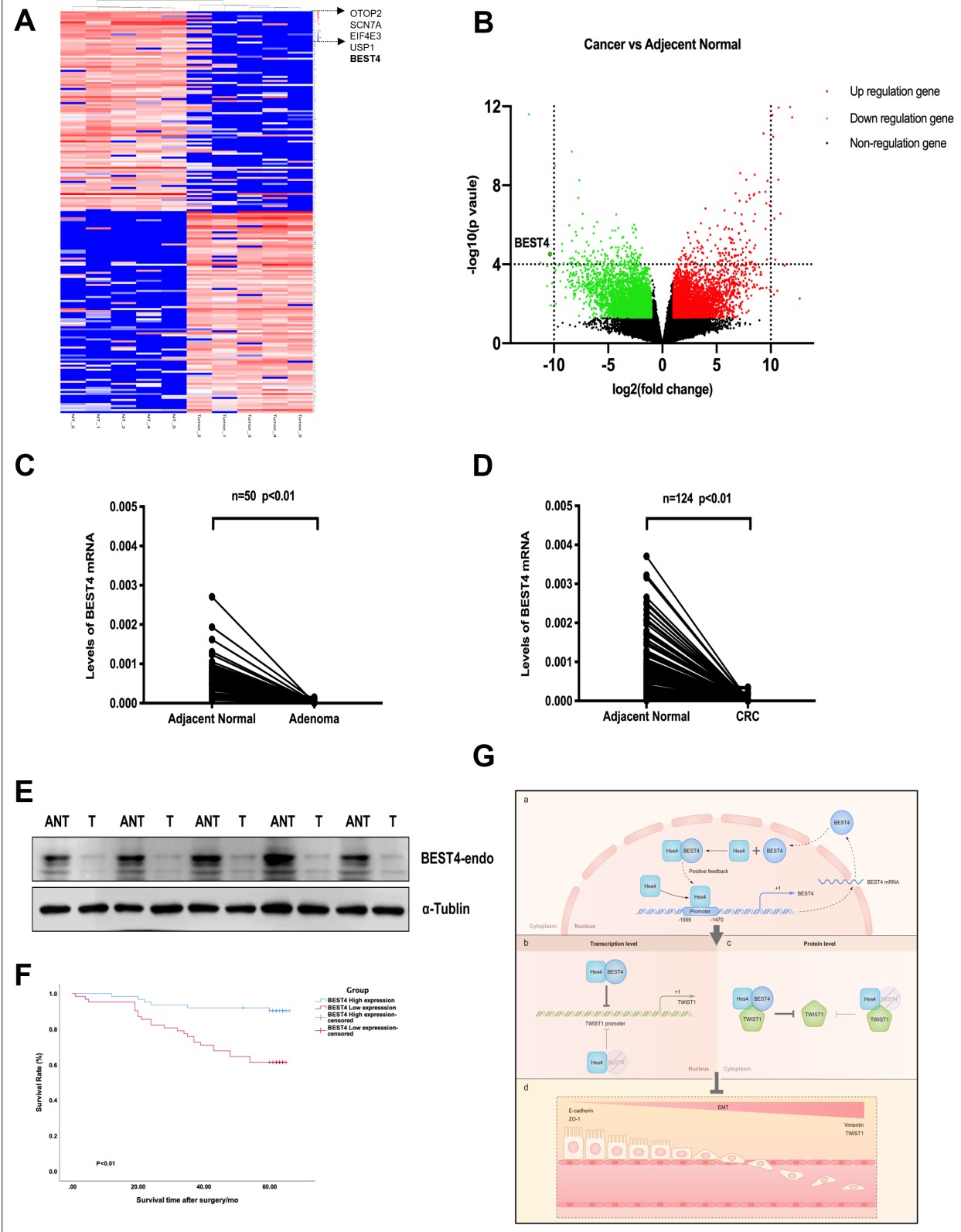

**Figure 5.** Correlation of the *BEST4* mRNA expression with clinicopathology and outcomes of patients diagnosed with CRC. (**A**) Heatmap based on the top 100 differentially expressed genes (downregulated and upregulated) in CRCs compared with ANTs. (**B**) Volcano plots of RNA-seq results showing all differentially expressed genes in CRC tissues compared with ANTs based on the fold-change (|FC|≥1) and p-value (p≤0.05). (**C**) Expression levels of *BEST4* mRNA in 50 pairs of adenoma tissues compared with ANTs. (**D**) Expression levels of *BEST4* mRNA in 124 pairs of CRC tissues compared

*Figure 5 continued on next page*

*Figure 5 continued*

with ANTs. (**E**) BEST4 protein levels in CRC and their matched counterparts were measured using western blotting. (**F**) Kaplan-Meier survival analysis according to *BEST4* expression in 124 patients with CRC. The difference was statistically significant based on the log-rank test (p<0.01). (**G**) A working model delineates the mechanism by which *BEST4* is transcriptionally activated by *HES4* and suppressed epithelial-to-mesenchymal transition (EMT) through TWIST1 inhibition: (a) the expression of *BEST4* is regulated by interactions of its upstream promoter region with HES4. *BEST4* positively promotes the transcriptional activity of HES4 on *BEST4* by physically binding to HES4 in a positive feedback loop. (b) The complex formed by BEST4 and HES4 interacts with TWIST1 and inhibits the promoter activity of TWIST1. HES4 alone is insufficient to inhibit TWIST1 promoter activity, and BEST4 is required. Inhibition of TWIST1 promoter activity by HES4 is BEST4-dependent. (c) The complex formed by BEST4 and HES4 destabilised TWIST1; the same effect caused by HES4 required BEST4 intermediation. The HES4-BEST4-TWIST1 axis counteracts EMT induction. *p<0.05, **p<0.01; CRC, colorectal cancer; ANT, adjacent normal tissue; T, tumour.

The online version of this article includes the following source data and figure supplement(s) for figure 5:

**Figure supplement 1.** *BEST4* suppresses epithelial-to-mesenchymal transition (EMT) independently of its channel functions.

**Figure supplement 1—source data 1.** Original files for western blot analysis displayed in *Figure 5—figure supplement 1D, E, and F*.

**Figure supplement 1—source data 2.** PDF file containing original western blots for *Figure 5—figure supplement 1D, E, and F*, indicating the relevant bands and treatments.

**Source data 1.** Original files for western blot analysis displayed in *Figure 5E*.

**Source data 2.** PDF file containing original western blots for *Figure 5E*, indicating the relevant bands and treatments.

HES4 and TWIST1 in the nucleus of cells, and protein-protein interactions; (3) the BEST4 inhibitory role in EMT exhibits upregulation of epithelial TJP1 and E-cadherin, and downregulation of mesenchymal VIM and TWIST1 in CRC; and finally (4) low *BEST4* mRNA levels were significantly correlated with malignant lesions and worse outcomes of CRC patients. To the best of our knowledge, this is the first characterisation of the role of BEST4 in suppressing CRC progression and the underlying molecular mechanisms.

CaCCs are expressed widely in several types of epithelial and endothelial cells and are classified as calcium-dependent chloride channels (e.g. anoctamin-1 [ANO1 or TMEM16]), chloride channel accessory proteins (CLCAs), and members of the BEST family. The latter is a recent addition to the class of molecularly defined chloride channels (*Huang et al., 2012*). Calcium-dependent chloride channels and CLCAs are differentially expressed in CRC and normal colon tissues, exerting pro- or anti-growth effects, while little is known of BEST family in CRC (*Li et al., 2017*; *Sanders et al., 2012*; *Sui et al., 2014*; *Yang et al., 2015*). There are four subtypes of the *BEST* family in mammals that assemble as pentamers to form $Ca^{2+}$-activated $Cl^-$ channels and maintain cellular physiological functions (*Marmorstein et al., 2009*; *Milenkovic et al., 2008*; *Tsunenari et al., 2003*). Subtypes of *BEST1-3* are responsible for vitelliform macular dystrophy, regulating the cilia of olfactory sensory neuronal activity; others regulate cGMP-dependent calcium-activated chloride conductance in vascular smooth muscle cells (*Matchkov et al., 2008*; *Pifferi et al., 2009*). An intrinsic function of the *BEST4* gene in modulating CRC growth remains poorly illustrated. In the present study, we demonstrated that BEST4 functionally inhibits the development of CRC in vitro and in vivo. BEST4 is transcriptionally regulated and physically interacts with upstream *HES4*, leading to downregulation of TWIST1 and EMT suppression. This suppression is independent of physiological $Ca^{2+}$-activated $Cl^-$ channels since inhibition of BEST4-associated $Ca^{2+}$- and $Cl^-$ fluxes did not alter TJP1/E-cadherin and TWIST1 protein expression.

The recent identification of a heterogeneous BEST4$^+$ and HES4$^+$ subgroup in a human colonic epithelial lineage (*Parikh et al., 2019*) led us to consider their potential role in regulating CRC progression. Among the HES family, which is associated with intestinal progenitor cells in adenoma transformation, HES1 and HES4 are a paralog pair, with other members more distantly related (*Katoh and Katoh, 2004*). While HES1 has been reported to induce EMT and promote metastasis and *chemoresistance* in CRC (*Sun et al., 2017*; *van Es et al., 2005*; *Yuan et al., 2015*), HES4 is identified by our present study capable of combating EMT induction in a BEST4-dependent manner that is regulated at both transcriptional and translational levels.

The process of EMT transforms epithelial cells exhibiting a spindle fibroblast-like morphology, leading to the acquisition of mesenchymal characteristics and morphology, enabling these cells to acquire invasive and migratory abilities, with expression switching epithelial E-cadherin and TJP1 to mesenchymal VIM (*Dongre and Weinberg, 2019*). When diagnosed in advanced stages, EMT may occur as CRC metastasize to distal organs (*Pastushenko and Blanpain, 2019*; *Sunlin Yong et al., 2021*; *Yeung and Yang, 2017*; *Zhang et al., 2021*). The whole process is regulated by transcriptional

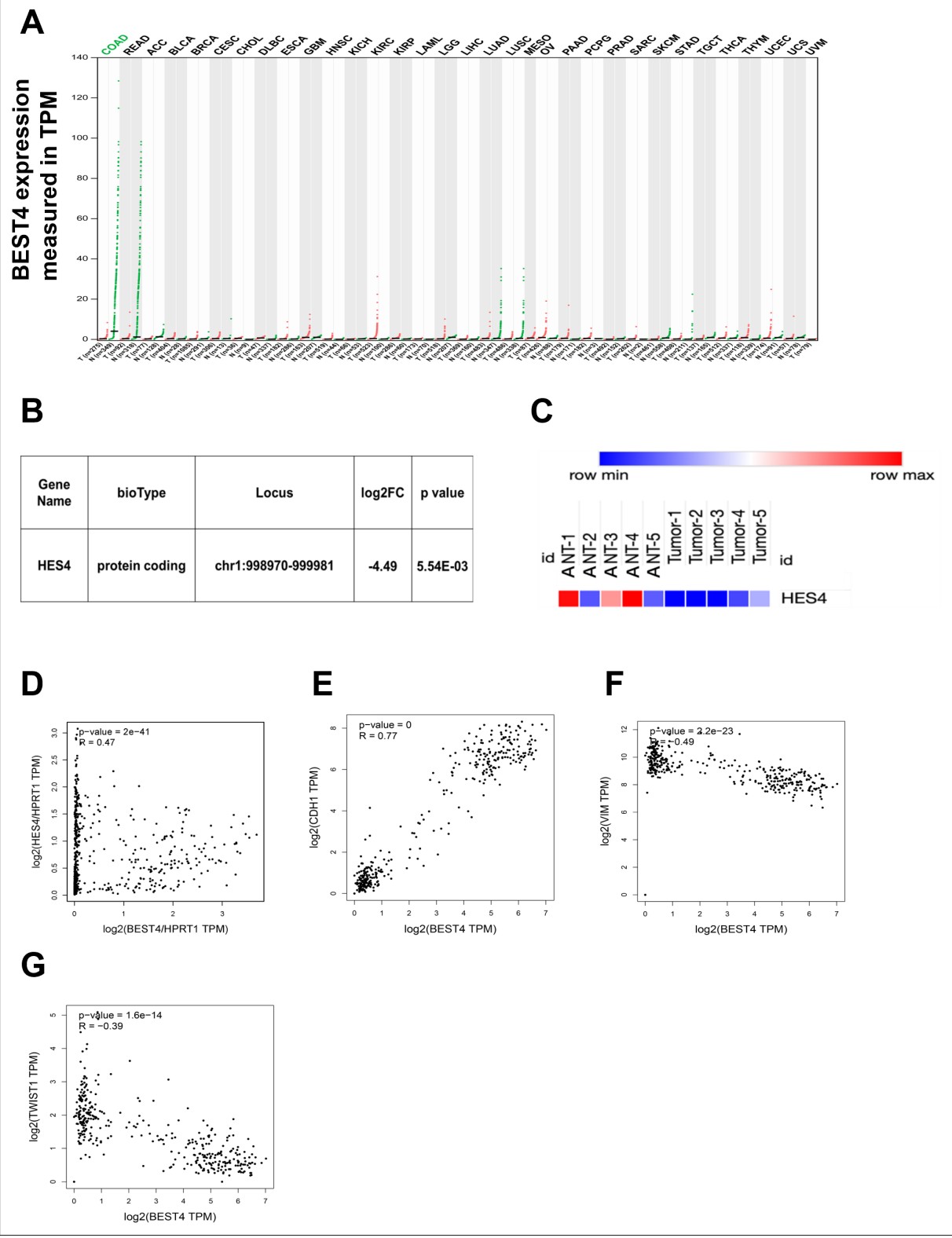

**Figure 6.** BEST4 mRNA expression in colorectal tumours and matched non-tumour tissues and the clinical relationship between BEST4, HES4, and epithelial-to-mesenchymal transition (EMT) markers. (**A**) Correlation between BEST4 and CDH1 in normal and colorectal cancer (CRC) samples analysed from the The Cancer Genome Atlas (TCGA) and Genotype-Tissue Expression (GTEx) databases using the Gene Expression Profiling Interactive Analysis (GEPIA) tool (http://gepia.cancer-pku.cn/). (**B**) RNA-seq data revealed decreased expression of HES4 in CRC tissues compared with ANTs. (**C**) Heatmap of HES4 expression in CRC and ANTs. (**D**) The correlation between BEST4 and HES4, in CRC analysed from the TCGA and GTEx databases using GEPIA

*Figure 6 continued on next page*

*Figure 6 continued*

(http://gepia.cancer-pku.cn/). (**E**) Correlation between BEST4 and CDH1 in normal and CRC samples analysed from the TCGA and GTEx databases using the GEPIA tool (http://gepia.cancer-pku.cn/). (**F**) Correlation between BEST4 and VIMin normal and CRC samples. (**G**) Correlation between BEST4 and TWIST1 in normal and cancer colorectal samples. ANTs, adjacent normal tissues; COAD, colon adenocarcinoma; READ, rectum adenocarcinoma; other tumour abbreviations are available at GEPIA: http://gepia.cancer-pku.cn/.

factors of the Snail family and TWIST1 (***Dongre and Weinberg, 2019***). TWIST1 (a basic helix-loop-helix transcription factor) reprograms EMT by repressing the expression of E-cadherin and TJP1 (***Nagai et al., 2016***; ***Yang et al., 2004***) and simultaneously inducing several mesenchymal markers, typically VIM (***Bulzico et al., 2019***; ***Meng et al., 2018***; ***Nagai et al., 2016***; ***Yang et al., 2004***), which

**Table 1.** Correlation of *BEST4* mRNA level with clinicopathological factors.

| Parameter | n | *BEST4* expression | | p-Value |
| --- | --- | --- | --- | --- |
| | | Low | High | |
| Age (year) | | | | 0.66 |
| <60 | 58 | 30 | 28 | |
| ≥60 | 67 | 32 | 35 | |
| Sex | | | | **0.022*** |
| Male | 74 | 43 | 31 | |
| Female | 51 | 19 | 32 | |
| Tumour differentiation | | | | 0.258 |
| Poor | 10 | 3 | 7 | |
| Moderate | 114 | 59 | 55 | |
| Well | 1 | 0 | 1 | |
| Tumour stage (TNM)[†] | | | | |
| I | 23 | 10 | 13 | **0.02*** |
| II | 50 | 18 | 32 | |
| III | 43 | 27 | 16 | |
| IV | 9 | 7 | 2 | |
| Lymph node metastasis | | | | **0.032*** |
| N0 | 75 | 30 | 45 | |
| N1 | 41 | 26 | 15 | |
| N2 | 9 | 6 | 3 | |
| Distant metastasis | | | | 0.08 |
| Absent | 116 | 55 | 61 | |
| Present | 9 | 7 | 2 | |
| Location | | | | 0.114 |
| Colon | 32 | 12 | 20 | |
| Rectum | 93 | 50 | 43 | |
| CEA level | | | | 0.13 |
| Low | 63 | 27 | 26 | |
| High | 62 | 35 | 27 | |

*Statistically significant.
[†]Staging was performed according to the eighth edition of AJCC (***Amin, 2017***).

**Table 2.** Univariate and multivariate analysis of prognostic factors for overall survival of patients with colorectal cancer (CRC).

| Variable | Comparison | Univariate analysis, p (log-rank test) | Multivariate analysis | | |
|---|---|---|---|---|---|
| | | | HR | 95% CI | p |
| Age (year) | ≥60 vs <60 | 0.001* | 2.713 | 1.524–4.810 | 0.001* |
| Sex | Female vs male | 0.681 | NA | NA | NA |
| Clinical stage | II, III, IV vs I | 0.1 | NA | NA | NA |
| Tumour differentiation | Moderate, well vs poor | 0.08 | NA | NA | NA |
| Lymph node metastasis | N1, N2 vs N0 | <0.001* | 2.190 | 1.291–3.717 | 0.004* |
| Distant metastasis | Present vs absent | <0.001* | 2.185 | 1.103–4.328 | 0.025* |
| Location | Rectum vs colon | 0.22 | NA | NA | NA |
| CEA level | High vs low | 0.002* | 1.697 | 0.953–2.960 | 0.073 |
| *BEST4* level | High vs low | <0.001* | 0.285 | 0.157–0.606 | <0.001* |

*Statistically significant.
HR = hazard ratio. CI = confidence interval. NA = not applicable.

is a pivotal predictor of CRC progression (*Vesuna et al., 2008*; *Yang et al., 2004*; *Yusup et al., 2017*; *Zhu et al., 2015*). The TWIST1 transcriptional factor plays a crucial role in inducing EMT, which contributes to the malignant lesions (*Yang et al., 2004*). Its physical interaction with HES4 determines human bone marrow stromal/stem cell differentiation (*Cakouros et al., 2015*). TWIST1, a basic helix-loop-helix transcription factor, represses the expression of E-cadherin and TJP1 and induces several mesenchymal markers including vimentin, which is a pivotal predictor of CRC progression (*Nagai et al., 2016*; *Yang et al., 2004*; *Bulzico et al., 2019*; *Meng et al., 2018*; *Nagai et al., 2016*; *Vesuna et al., 2008*; *Yang et al., 2004*; *Yusup et al., 2017*; *Zhu et al., 2015*). Overexpression of TWIST1 significantly enhances the migration and invasion capabilities of CRC cells; furthermore, it is closely associated with metastasis and poor prognosis in patients with CRC (*Yusup et al., 2017*; *Zhu et al., 2015*). Despite TWIST1 importance in CRC, little is known about how it is controlled during EMT induction. Our present study reveals that TWIST1 colocalised and interacted with BEST4 in the nucleus of CRC HCT116 cells, leading to downregulation of TWIST1 and inhibition of EMT. HES4 also caused the same effect, but through intermediation with BEST4. Although the mechanistic insights of their interplay remain to be elucidated, the present study identifies the HES4-BEST4-TWIST1 axis that contributes to counter EMT in the development of CRC (*Figure 5G*).

Lastly, our analysis of correlations reveals the clinical significance of BEST4 expression in CRC. In particular, the mRNA levels of BEST4 were lower in tissues of both precancerous adenomatous lesions and CRC than ANTs. These alterations are significantly correlated with large tumour size, positive nodes, and poor overall survival, especially in male patients. Interestingly, a previous study conducted on a Chinese cohort reported an approximately 40% higher incidence of CRC in males than in females (*Zhang et al., 2019*). But it remains to be investigated whether sex hormone-related factors play a role in the gender-specific downregulation of BEST4 and its impact on CRC in males (*Yu et al., 2020*).

In conclusion, the functional significance of BEST4 in CRC development and its interplay with other molecules is completely unknown. The present study demonstrates the role of BEST4 in inhibiting CRC proliferation, migration, and invasion both in models of in vitro and in vivo. Mechanically, *BEST4* is epistatic to HES4 for downregulating TWIST1 and inhibits EMT in CRC. Therefore, we reveals a previously unknown mechanism of the HES4-BEST4-TWIST1 axis that restrains the progression of CRC.

## Materials and methods
### Clinical samples
A total of 124 samples of paired CRC and ANTs and 50 samples of paired adenoma and ANT were collected randomly from clinical specimens from patients undergoing intestinal resections or upper

gastrointestinal endoscopies at the Department of Gastrointestinal Surgery and Digestive Endoscopy Centre, West China Hospital, China, between January and December 2013. All patients were followed until December 2018 or until death. Samples from patients with malignant tumours in other organs, from patients who received preoperative chemotherapy or radiotherapy, and from patients with incomplete follow-up data were excluded. The survival time was calculated from the surgery date to the follow-up deadline (above) or until the date of death. The Ethics Committee of West China Hospital, Sichuan University (Sichuan, China) approved the study. All procedures were performed per the 1964 Declaration of Helsinki. Written informed consent was obtained from all participants.

## Cell cultures and reagents

The human colon cancer cell lines Caco-2, HCT116, HCT-15, and LS174T were purchased from The American Type Culture Collection (Manassas, VA, USA). Cells were authenticated using short tandem repeat analysis performed within 2 years of receipt. Cells at passages>15 were not used. All cells were maintained in Dulbecco's Modified Eagle's medium (DMEM; Gibco, Waltham, MA, USA) supplemented with 10% foetal bovine serum (FBS), 100 U/mL penicillin, and 100 mg/mL streptomycin. Before each experiment, cells were provided with low glucose (1 g/L) DMEM with 5% FBS before adding experimental reagents. All cell lines were frequently examined using the MycoAlert Mycoplasma Detection Kit (Lonza, Verviers, Belgium) to ensure they remained free of mycoplasma contamination.

## Tumour and CRC hepatic metastasis mouse xenografts

Four-week-old female BALB/c nude mice were housed under standard conditions. The mice were randomised into two groups, with eight mice per group. HCT116 cells ($5 \times 10^6$ cells in 0.1 mL phosphate-buffered saline) transduced with the lentivirus containing *BEST4* or EV were injected subcutaneously into the right dorsal flank of mice. Tumour growth was measured twice a week using callipers, and tumour volumes were calculated as length × width$^2$ × 0.5. Mice were sacrificed at 32 days, and tumour weight was measured at the end of the experiment. For further analysis, the excised tumour tissues were divided into two parts and embedded in paraffin or preserved at −80°C for further analysis.

For the liver metastasis of CRC, intrasplenic injections of HCT-15 cells with BEST4-knockout (BEST4-gRNA), BEST4-rescued or control (EV-gRNA) ($5 \times 10^6$ cells in 0.1 mL PBS) were performed into 6-week-old Balb/c nude mice as described previously. There were 3 mice/group. Briefly, at day 28 after the procedure, mouse livers were paraffin embedded and the sections for H&E staining. The numbers of the metastatic nodules on the livers were double-blind visually quantified and confirmed by a pathologist.

The Animal Ethics Committee approved animal experiments of West China Hospital, Sichuan University (Sichuan, China).

## CRC cell lines with overexpression of *BEST4*

Forced overexpression of *BEST4* was examined in HCT116 and Caco2 CRC cells in which the C-terminal HA epitope-tagged human *BEST4* gene with GFP (*BEST4*-HA-GFP) or GFP alone was introduced by lentivirus-mediated gene transfer. Fluorescence-activated cell sorting (FACS) was performed to obtain single cells distributed into 96-well plates. Three independent lines with the highest levels of BEST4 expression were used for subsequent experiments (*Figure 1—figure supplement 1A and B*). *BEST4* mRNA and HA-tagged protein levels were significantly higher in HCT116 and Caco2 cells transfected with *BEST4*-HA plasmids than those transfected with the EV alone (*Figure 1—figure supplement 1C*). The *BEST4*-overexpressing cell lines were named 'HCT116 *BEST4*-HA' and 'Caco2 *BEST4*-HA'. Cells transfected with the EV alone were referred to as 'HCT116 EV' and 'Caco2 EV'.

## CRISPR/Cas9-mediated knockout of *BEST4*

*BEST4*-depleted HCT-15 cells were generated with CRISPR/Cas9 gene-editing technology using a dual sgRNA approach. Two sequences that targeted exon 3 within the *BEST4* active fragment (5'-TGACTCTCGTGGTGAACCGC-3' and 5'-CCGAGATGACGCACATCAGC -3') were inserted into the pU6-gRNA-Cas9-EGFP vector (GenePharma, Shanghai, China) to generate a CRISPR/Cas9 *BEST4* knockout plasmid. Excision of the targeted 44 bp genomic fragment resulted in splicing errors in exons 2–4 and generated a premature STOP codon that abrogated functional gene expression (*Figure 1—figure supplement 1F and G*). Cells were transfected with these plasmids, and GFP⁺ cells

were FACS-sorted as single cells into 96-well plates. After culturing for 3 weeks, cells were distributed into two 24-well plates, and the appropriate deletion was verified by sequencing. Only cell lines with homozygous deletions were used. Three independent lines generated using this method were used for subsequent experiments and were named 'HCT15 *BEST4*-gRNA'. Diminished expression of immunoreactive BEST4 protein was confirmed by immunoblotting (*Figure 1—figure supplement 1H*). The same overexpression strategy was adopted to rescue BEST4 expression in *BEST4* knockout HCT-15 cells, named 'HCT15 *BEST4*-Rescued'.

## Plasmids and transfection/transduction

Expression plasmids encoding *BEST4* (GeneID: 266675, NM_153274; vector: pCDH-CMV-MCS-EF1-CopGFP-T2A-Puro) containing a C-terminal hemagglutinin (HA) epitope tag, *HES4* (GeneID: 57801, NM_001142467.2; vector: pCDH-CMV-MCS-EF1- CopGFP-T2A-Puro) containing an N-terminal Flag epitope tag, were synthesised by Public Protein/Plasmid Library (Nanjing, China). According to the manufacturer's instructions, Lipofectamine 3000 (Invitrogen, Carlsbad, CA, USA) was used in cell transfection experiments.

To generate a model of CRC with forced overexpression of *HES4*, cells were transfected with lentiviral pCDH-CMV-MCS-EF1-CopGFP-T2A-Puro vectors containing human HES4 cDNA and selected with puromycin (Invivogen, San Diego, CA, USA) at 10 μg/mL introduced 48 hr after transfection.

For shRNA and siRNA experiments, HEK293 cells were co-transduced with lentiviral vectors carrying shRNA targeting *HES4* (Public Protein/Plasmid Library, Nanjing, China; target sequences are listed in *Supplementary file 1*). Supernatants were collected at 48 and 72 hr for analysis. Scrambled shRNAs were used to control for non-specific effects. Transduction was performed by mixing aliquots of lentivirus with a standard medium containing 10 μg/mL puromycin (Invivogen) to remove untransduced cells. GeneSolution siRNAs targeting *BEST4* (*BEST4*-siRNA) were used to silence gene expression. Scrambled siRNA (SCR) was used as a control (QIAGEN, Hilden, Germany). The target sequences of the *BEST4* siRNAs are listed in *Supplementary file 1*. For siRNA experiments, cells were seeded 1 day before transfection in an antibiotic-free medium and grown to 60% confluency in six-well plates. Cells were transfected using Lipofectamine 3000 as per the manufacturer's protocol. The efficiency of RNA interference was determined using qPCR and western blotting.

## RNA isolation, cDNA library preparation, sequencing, and data analysis

Total RNA was isolated using a High Pure RNA Isolation Kit (Roche, Basel, Switzerland). RNA samples with an RNA integrity number (RIN)≥7.0 and a 28S:18S ratio≥1.5 were used in subsequent experiments. Sequencing libraries were generated and sequenced by CapitalBio Technology (Beijing, China). A total amount of 3 μg RNA per sample was used. Briefly, the Ribo-Zero Magnetic Kit (Epicentre) was used to remove rRNA from the total RNA, and then the NEBNext Ultra RNA Library Prep Kit for Illumina (NEB, USA) was used to construct the libraries for sequencing following the manufacturer's instructions. The RNA was fragmented into pieces of ~200 base pairs in length in NEBNext First-Strand Synthesis Reaction Buffer (5×). The first-strand cDNA was synthesised from the RNA fragments using reverse transcriptase and random hexamer primers, and then the second-strand cDNA was synthesised in Second-Strand Synthesis Reaction Buffer with dUTP Mix (10×). The end of the cDNA fragment was subjected to an end-repair process that included the addition of a single 'A' base followed by ligation of the adapters. Products were purified and enriched using PCR to amplify the library DNA. The final libraries were qualified using Agilent 2100 and quantified using the KAPA Library Quantification Kit (KAPA Biosystems, South Africa). Finally, the libraries were subjected to paired-end sequencing with a paired-end 150-base pair reading length on an Illumina NovaSeq sequencer (Illumina).

The sequencing quality of raw data in *fastq* format was assessed using FastQC, and then low-quality data were filtered using the Next Generation Sequencing Quality Control platform. The high-quality clean reads were then aligned to the reference genome using Tophat2 with default parameters. The genome of the human genome version of hg38 (UCSC) was used as the reference. Cufflinks and Cuffmerge software packages were used to assemble the transcripts. All subsequent comparative analyses of lncRNAs and mRNAs were based on the results of the transcripts. Transcripts corresponding to mRNAs or lncRNAs were collected from databases and treated as known lncRNAs and annotated based on information from public databases. The transcripts were treated as novel lncRNAs when their sequence length was >200 and when they were predicted to be non-coding

RNAs. Differential expression analyses were performed using the limma package and edgeR. Functional annotation and enrichment analyses were performed using KOBAS 3.0. The target genes of the mRNAs were predicted based on cis- and trans-patterns based on the location of the reference genome and sequence similarity. The raw sequence data reported in this paper were deposited in the NCBI Sequence Read Archive (SRA) with accession number PRJNA690126.

### RNA extraction and real-time PCR

Total RNA was isolated from cell lysates according to the manufacturer's protocol (Zymo Research, Tustin, CA, USA). A High-Capacity cDNA Reverse Transcription Kit (Thermo Fisher Scientific, Waltham, MA, USA) was used for first-strand cDNA synthesis with 1 μg of total RNA from each sample. The expression of genes and the GAPDH internal control was assayed by qPCR using SYBR Green (Bioline Reagents Ltd, London, UK). Primer sequences are shown in *Supplementary file 1*.

### Immunoblotting

Protein extracts were prepared using RIPA lysis buffer (Merck Millipore, Temecula, CA, USA) containing protease inhibitor cocktail and phosphatase inhibitor tablets (Sigma, MO, USA). Protein content was determined using a protein assay kit (Bio-Rad, Hercules, CA, USA). Western blotting was performed as described previously (*Wang et al., 2019*). All primary (1:1000) and secondary antibodies (1:2000), except BEST4 (1:800; LsBio, Seattle, WA, USA) and anti-Flag M2 (Sigma, MO, USA), were purchased from Cell Signaling Technology (Danvers, MA, USA).

### Cell proliferation

Cell proliferation was monitored in real time using an *IncuCyte* ZOOM live-cell microscope incubator (Essen Bioscience, MI, USA), in which the cell confluence was calculated using *IncuCyte* software, and the cell proliferation was expressed as the percentage of confluence. According to the manufacturer's protocol, a Cell Counting Kit-8 (CCK-8, Dojindo, Tokyo, Japan) assay was used to confirm cell viability. The absorbance at 450 nm was measured using a microplate reader (BioTek, VT, USA), and growth curves were drawn according to the optical density.

### Clonogenesis

A total of $3×10^3$ cells per well were seeded in six-well plates. The medium was refreshed twice a week. After 2 weeks, colonies were fixed and stained with crystal violet (0.005% in 20% methanol). The numbers of colonies were quantified using GelCount (Oxford Optronix Ltd, Oxford, UK). The assay was carried out in triplicate wells for three independent experiments.

### Transwell migration and invasion

Cell migration and invasion assays were performed using Transwell assay with uncoated polycarbonate transwell inserts (Millipore, Billerica, MA, USA). Cells were digested and suspended in culture medium without FBS for the migration assay. Cell suspensions were added into the upper chamber of the transwell insert and then placed into the 24-well plate transwell containing medium with 10% FBS in the lower chamber. For the invasion assay, the 24-well plate with transwell inserts was coated with 50 μg/insert of Matrigel matrix on the top of the chamber (BD Biosciences, Oxford, UK). After 48 hr, cells that travelled to the lower surface of the insert were stained with crystal violet. Experiments were conducted in triplicate.

### Immunohistochemistry

Paraffin-embedded tissue blocks from formalin-fixed tumour samples were sectioned, dewaxed, and rehydrated following standard protocols. Sections were stained for human Ki67 primary antibody (1:200, Cell Signaling Technology, MA, USA), a marker of cell proliferation, and anti-mouse HRP-conjugated secondary antibody (Dako, CA, USA), and the positive cells were counted in eight randomly chosen fields. Expression of BEST4-HA in tumours was stained with anti-HA primary antibody (1:800, Cell Signaling Technology) and anti-rabbit HRP-conjugated secondary antibody (Dako).

### Dual-luciferase reporter assay

The *BEST4* promoter region (−2000 to +100 bp) was cloned into pGL3-basic vectors (Promega, WI, USA) to construct the luciferase reporter plasmid (pGL3-*BEST4*). The stably high HES4 expression

cell line was constructed by infecting HCT116 cells with lentivirus, and then the recombinant plasmid pGL3- *BEST4* was co-transfected with the control plasmid pRL-TK (Promega) into HCT116 HES4 and HCT116 EV cells. Twenty-four hours after transfection, firefly and Renilla luciferase activities were analysed using the dual-luciferase reporter assay (Promega) according to the manufacturer's instructions, and the results were normalised as relative luciferase activity (firefly luciferase/Renilla luciferase).

## Chromatin immunoprecipitation-quantitative PCR

ChIP-qPCR was performed as previously described (*Miotto and Struhl, 2006*). At room temperature, we crosslinked cells at $5 \times 10^7$ cells/mL for 10 min using 1% formaldehyde and stopped the reaction using 125 mM glycine. After three washes with ice-cold PBS, cells were homogenised, lysed, and sonicated to shear DNA into 200–500 bp fragments. The DNA fragments were incubated with 10 μg of antibodies against anti-Flag tag (Sigma, MO, USA) or native IgG (Cell Signaling Technology) followed by IP with 60 μL of protein A/G Magnetic Beads (MedChemExpress, NJ, USA) and overnight incubation at 4°C with rotation. Immunoprecipitated DNAs were extracted from the DNA/antibody/protein A/G bead complexes using proteinase K digestion, subjected to reverse crosslinking at 65°C for 4 hr, and purified with spin columns. qPCR analysis was performed to determine the enrichment of DNA fragments using the indicated primers (*Supplementary file 1*). ChIP-qPCR data were normalised using the fold-enrichment method (ChIP signals were divided by IgG signals), and DHFR 5 'UTR was used as a negative control. Each experiment included triplicate samples and the data shown represent the mean of three independent experiments.

## Immunofluorescence staining

HCT116 cells transfected alone or co-transfected with expression vectors encoding C-terminal HA-tagged BEST4, and N-terminal Flag-tagged HES4 were grown on glass coverslips, fixed for 10 min with 4% paraformaldehyde, permeabilised for 5 min with 0.2% Triton X-100, then blocked for 15 min with 3% bovine serum albumin. HA-tagged *BEST4* was stained using an anti-HA antibody (Cell Signaling Technology) followed by a TRITC-labelled secondary antibody (Abcam, Cambridge, UK). Flag-tagged HES4 was detected using an anti-Flag antibody (Sigma, MO, USA) with an Alexa Fluor 647-labelled secondary antibody (Invitrogen, Carlsbad, CA). Cells were mounted and imaged using laser-scanning confocal microscopy (Nikon Corporation, Tokyo, Japan).

## Co-immunoprecipitation

Co-IP assays were performed on nuclear extracts. According to the manufacturer's protocols, nuclear proteins of each sample were collected using Minute Cytoplasmic and Nuclear Fractionation Kit (Invent, MN, USA). Protein G Dynabeads (Thermo Fisher Scientific, MA, USA) following the manufacturer's instructions. Briefly, HEK293 cells were co-transfected with expression vectors encoding C-terminal HA-tagged *BEST4* and N-terminal Flag-tagged HES4. Cells were harvested and lysed 48 hr after transfection. Cell lysates (500 μg) were incubated overnight at 4°C in a rotating wheel with 2 μg of anti-FLAG (Sigma, MO, USA) or anti-HA antibody (Cell Signaling Technology) bound to Protein G Dynabeads. Immunocomplexes were separated using a magnet, washed three times in PBS with 0.02% Tween 20 and eluted in elution buffer (50 mM glycine pH 2.8). After boiling in sample buffer at 70°C for 10 min, immunocomplexes were loaded on 4–20% Tris-Glycine gels (Bio-Rad, CA, USA), followed by blotting on PVDF membranes and immunoblotting using the indicated antibodies.

## Chloride channel efflux

Cells were maintained in a serum-free DMEM overnight prior to treatment with DIDS (4,4'-diisothiocyanatostilbene-2,2'-disulfonic acid; HY-D0086, MCE, China), CaCC$_{inh}$-A01 (HY-100611, MCE, China), or 1 μM ionomycin for 24 hr as previously published (*Liu et al., 2015*; *Uwada et al., 2019*). Chloride channel efflux was evaluated with colorimetric changes by commercially available kits (ab176767, Abcam). Briefly, 40,000 *BEST4*-expressing or *EV*-control HCT116 cells were plated per well in a 96-well plate and allowed to grow overnight. Before the assay, cells were refreshed with serum-free medium and 100 μL/well of pre-warmed iodide-loading buffer was added and the plate was incubated for 2 hr at 37°C. After washing each well with HBSS three times, cells were lysed and Iodide Sensor Blue dye (50 μL/well) and 0.5× Iodide Sensor Enhancer (50 μL/well) were added and incubated for 1 min before reading at 405 nm. Data was calculated as fold-change relative to control.

## Statistical analysis

The results were expressed as the mean ± standard error of the mean. The statistical significance of differences between means was assessed using Student's unpaired two-tailed t-tests, and the Mann-Whitney U test was performed to compare the variables in two groups using SPSS 25.0 (SPSS, Chicago, IL, USA). The relationship between the clinicopathological parameters of CRC and the expression of *BEST4* was examined using the Kruskal-Wallis H test. The Kaplan-Meier method and log-rank tests were used to assess overall survival in relation to *BEST4* expression. Differences in cell proliferation and tumour growth rates between the two groups of mice were assessed using repeated-measures analysis of variance. Western blot intensity was assessed using ImageJ v1.53 software (ImageJ, Bethesda, MD, USA). p-Values<0.05 indicated statistically significant differences.

## Acknowledgements

We thank Xuemei Cheng, Sisi Wu, and Huifang Li from Core Facilities of West China Hospital, Sichuan University, for expert technical assistance. This work was supported by the National Natural Science Foundation of China (grant numbers: 82173253, 82103539, 82074298), Sichuan Science and Technology Program (grant numbers: 2021YFH0005, 2022YFH0003), Sichuan Province cadres health research project (grant numbers: 2023104).

## Additional information

### Funding

| Funder | Grant reference number | Author |
|---|---|---|
| National Natural Science Foundation of China | 82103539 | Zijing Wang |
| National Natural Science Foundation of China | 82074298 | Ziyi Zhao |
| Sichuan Association for Science and Technology | 2021YFH0005 | Zijing Wang |
| Sichuan Association for Science and Technology | 2022YFH0003 | Jilin Yang |
| Health Commission of Sichuan Province | 2023104 | Zijing Wang |
| National Natural Science Foundation of China | 82173253 | Jilin Yang |

The funders had no role in study design, data collection and interpretation, or the decision to submit the work for publication.

### Author contributions

Zijing Wang, Conceptualization, Funding acquisition, Investigation, Writing – original draft, Project administration, Writing – review and editing; Bihan Xia, Xian Zhang, Miao Zhang, Software, Methodology; Shaochong Qi, Software, Writing – original draft; Xiaoshuang Zhang, Investigation; Yan Li, Huimin Wang, Methodology; Ziyi Zhao, Software, Methodology, Writing – original draft; David Kerr, Li Yang, Supervision, Funding acquisition, Writing – review and editing; Shijie Cai, Conceptualization, Supervision, Funding acquisition, Writing – review and editing; Jilin Yang, Conceptualization, Supervision, Funding acquisition, Project administration, Writing – review and editing

### Author ORCIDs

Zijing Wang 
Shijie Cai 
Jilin Yang 

## Ethics

Human subjects: The Ethics Committee of West China Hospital, Sichuan University (Sichuan, China) approved the study. All procedures were performed per the 1964 Declaration of Helsinki. Written informed consent was obtained from all participants.

This study was performed in strict accordance with the recommendations in the Guide for the Care and Use of Laboratory Animals of the National Institutes of Health. All animal experiments were approved by the Institutional Animal Care and Treatment Committee of Sichuan University in China (New Permit Number: 20131205010). All surgery was performed under sodium pentobarbital anesthesia, and every effort was made to minimize suffering.

Reviewer #1 (Public review): https://doi.org/10.7554/eLife.88879.3.sa1
Reviewer #2 (Public review): https://doi.org/10.7554/eLife.88879.3.sa2
Author response https://doi.org/10.7554/eLife.88879.3.sa3

---

# Additional files

## Supplementary files

• Supplementary file 1. Table S1. The target sequence of short hairpin RNAs (shRNAs), RNAs silencing (siRNAs), and primers used in this study.

• MDAR checklist

## Data availability

The raw sequence data reported in this paper have been deposited in the NCBI Sequence Read Archive (SRA) with accession number PRJNA690126.

The following dataset was generated:

| Author(s) | Year | Dataset title | Dataset URL | Database and Identifier |
|-----------|------|---------------|-------------|-------------------------|
| Wang Z | 2022 | Expression of Cystatin SN and DPT and their mechanisms in the development of colorectal cancer(CRC) | http://www.ncbi.nlm.nih.gov/bioproject/?term=PRJNA690126 | NCBI BioProject, PRJNA690126 |

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
