## [Editor Report · eLife Assessment]

The findings of this **valuable** manuscript advance our understanding of the significance of Bestrophin isoform 4 (BEST4) in suppressing colorectal cancer (CRC) progression. The authors used appropriate and validated methodology, such as the knockout of BEST4 using CRISPR/Cas9 in CRC cells, to provide a **solid** foundation for elucidating the potential link between BEST4 and CRC progression.

---

## [Referee Report · Reviewer #1 (Public review)]

Summary:

In this study, the authors describe the participation of the Hes4-BEST4-Twist axis in controlling the process of epithelial-mesenchymal transition (EMT) and the advancement of colorectal cancers (CRC). They assert that this axis diminishes the EMT capabilities of CRC cells through a variety of molecular mechanisms. Additionally, they propose that reduced BEST4 expression within tumor cells might serve as an indicator of an adverse prognosis for individuals with CRC.

The revised manuscript and figures still need further improvement because some of the authors' claims are difficult to understand from a scientific perspective.

Strengths:

• Exploring the correlation between the Hes4-BEST4-Twist1 axis, EMT, and the advancement of CRC is a novel perspective and gives readers a fresh standpoint.

• The potential role of BEST4 in EMT through the Hes4-BEST4-Twist1 axis, rather than through its channel function, is also a novel perspective.

• The whole transcriptome sequence analysis (Figure 5) showing low expression of BEST4 in CRC samples will be of broad interest to cancer specialists as well as cell biologists although further corroborative data is essential to strengthen these findings (See Weaknesses).

Weaknesses:

• The authors employed three kinds of CRC cell lines, but not untransformed cells such as intestinal epithelial organoids which are commonly used in recent research. Since all the data from in vitro and in vivo experiments are generated from CRC cell lines with forced expression of proteins of interest, the authors' claim may not reflect a common biological process.

• Most of experiments were performed to show changes in EMT markers, but not EMT itself.

• The in vivo and in vitro data supporting the whole transcriptome sequence analysis (Figure 5) is mostly insufficient. Since BEST4 is a marker of a subset of terminally differentiated colonocytes, its lower expression in CRC compared to adjacent normal tissue could be within the range of common expectations.

• Some experiments do not appear to have a direct relevance to their claims.

Major comments:

• The authors employed three kinds of CRC cell lines, but not untransformed cells such as intestinal epithelial organoids which are commonly used in recent research. Please include this limitation of the study in the discussion section with other possible limitations.

• Some experiments do not appear to have direct relevance to their claims. Figure 1A-1F and 2E-2H relate to cell proliferation or viability of CRC cell lines, but not to EMT. The focus of this study should be on EMT, but the summary sentence for Figure 1 (Line 118-119) says "inhibitory effects of BEST4 on CRC development". This sentence, along with some others (such as Line 262-263), seems to be deviating. Cancer development and EMT are distinct biological processes, so please revise the manuscript with this in mind.

• The context around Line 194-197, "Additionally, the knockdown of endogenous BEST4 in Hes4-expressing HCT116 cells substantially decreased Flag-Hes4 coprecipitation from the nuclear protein lysates, while Myc-Twist1 expression remained constant, as determined by co-IP with antibodies to Flag or My (Figure 4E; Figure 4-figure supplement 1C)." is difficult to follow.

• The in vivo and in vitro data supporting the whole transcriptome sequence analysis (Figure 5) are mostly insufficient. Since BEST4 is a marker of a subset of terminally differentiated colonocytes, its lower expression in CRC compared to adjacent normal tissue could be within the range of common expectations.

• As the reviewer #2 mentioned, the quality of some figures is quite suboptimal. It is not due to their pixel size, but rather due to other factors, such as the inconsistencies of aspect ratios. Improvement of the overall quality is needed. Figure legends also need improvements.

• The formatting of genes and proteins is inconsistent. Please correct it according to the general formatting guidelines.

---

## [Referee Report · Reviewer #2 (Public review)]

Summary:

Using in vitro and in vivo approaches, the authors first demonstrate that BEST4 inhibits intestinal tumor cell growth and reduces their metastatic potential, possibly via downstream regulation of TWIST1.

They further show that HES4 positively upregulates BEST4 expression, with direct interaction with BEST4 promoter region and protein. The authors further expand on this with results showing that negative regulation of TWIST1 by HES4 requires BEST4 protein, with BEST4 required for TWIST1 association with HES4. Reduction of BEST1 expression was shown in CRC tumor samples, with correlation of BEST4 mRNA levels with different clinicopathological factors such as sex, tumor stage and lymph node metastasis, suggesting a tumor-suppressive role of BEST4 for intestinal cancer.

Strengths:

• Good quality western blot data

• Multiple approaches were used to validate the findings

• Logical experimental progression for readability

• Human patient data / In vivo murine model / Multiple cell lines were used, which supports translatability/reproducibility of findings

Weaknesses:

• Figure quality should still be improved

• The discussion should still be improved

---

## [Author Response]

The following is the authors’ response to the original reviews.

**Public Reviews:**

**Reviewer #1 (Public Review):**
Summary:In this study, the authors describe the participation of the Hes4-BEST4-Twist axis in controlling the process of epithelial-mesenchymal transition (EMT) and the advancement of colorectal cancers (CRC). They assert that this axis diminishes the EMT capabilities of CRC cells through a variety of molecular mechanisms. Additionally, they propose that reduced BEST4 expression within tumor cells might serve as an indicator of an adverse prognosis for individuals with CRC.Strengths:• Exploring the correlation between the Hes4-BEST4-Twist axis, EMT, and the advancement of CRC is a novel perspective and gives readers a fresh standpoint.• The whole transcriptome sequence analysis (Figure 5) showing low expression of BEST4 in CRC samples will be of broad interest to cancer specialists as well as cell biologists although further corroborative data is essential to strengthen these findings (See Weaknesses).Weaknesses:(1) The authors employed three kinds of CRC cell lines, but not untransformed cells such as intestinal epithelial organoids which are commonly used in recent research.

Sincerely thanks for catching this issue. While we acknowledge the potential of intestinal epithelial organoids as a valuable model for this study and will consider establishing this system in future research, which falls outside the scope of our current work.

(2) The authors use three different human CRC cell lines with a lack of consistency in the selection of them. Please clarify (1) how these lines are different from each other, (2) why they pick up one or two of them for each experiment. To be more convincing, at least two lines should be employed for each in vitro experiment.

We apologize for any confusion caused to the reviewer. In our study, we employed HCT116 and Caco2 cell lines to investigate the overexpression of BEST4 in the biological functions of CRC and its involvement in EMT. The selection of HCT116, a human CRC cell line, was based on its relatively lower expression level of BEST4 compared to other CRC cell lines. Conversely, Caco2 is a human colon adenocarcinoma cell line that closely resembles differentiated intestinal epithelial cells and exhibits microvilli structures. Given that BEST4 serves as a marker for intestinal epithelial cells, these two cell lines were chosen for investigating the in vitro effects of overexpressing BEST4 on proliferation, clonality, invasion, migration of colon cancer tumor cells and expression of downstream EMT-related markers. Similarly, we selected the HCT-15 cell line derived from human CRC for BEST4 knockout due to its comparatively higher expression level of BEST4 among other CRC cell lines. We employed the CRISPR/Cas9 gene-editing technology to knockout BEST4 instead of utilizing shRNA for downregulating BEST4 expression, thereby limiting our selection to a single cell line.

(3) The authors demonstrated associations between BEST4 and cell proliferation/ viability as well as migration/invasion, utilizing CRC cell lines, but it should be noted that these findings do not indicate a tumor-suppressive role of BEST4 as mentioned in line 120. Furthermore, while the authors propose that "BEST4 functions as a tumor suppressor in CRC" in line 50, there seems no supporting data to suggest BEST4 as a tumor suppressor gene.

We apologize for these inaccurate expressions, and we have made the necessary modifications to the corresponding parts in the manuscript.

(4) The HES4-BEST4-Twist1 axis likely plays a significant role in CRC progression via EMT but not CRC initiation. Some sentences could lead to a misunderstanding that the axis is important for CRC initiation.

We apologize for these inaccurate expressions, and we have made the necessary modifications to the corresponding parts in the manuscript.

(5) The authors mostly focus on the relationship of the HES4-BEST4-Twist1 axis with EMT, but their claims sometimes appear to deviate from this focus.

We apologize for confusing the reviewer. The objectives of our study are as follows: (1) to establish the role of BEST4 in CRC growth both in vitro and in vivo; (2) to determine the underlying molecular mechanisms by which BEST4 interacts with Hes4 and Twist1, thereby regulating EMT; and (3) to investigate the correlation between BEST4 expression and prognosis of CRC. We have made the necessary modifications to the corresponding parts in the manuscript.

(6) Some experiments do not appear to have a direct relevance to their claims. For example, the analysis using the xenograft model in Figure 2E-J is not optimal for analyzing EMT. The authors should analyze metastatic or invasive properties of the transplanted tumors if they intend to provide some supporting evidence for their claims.

Sincerely thanks for catching this issue. The process of EMT transforms epithelial cells exhibiting a spindle fibroblast-like morphology, leading to the acquisition of mesenchymal characteristics and morphology, enabling these cells to acquire invasive and migratory abilities, with expression switching epithelial E-cadherin and Zo-1 to mesenchymal vimentin (Dongre and Weinberg, 2019)..The whole process is regulated by transcriptional factors of the Snail family and Twist1(Dongre and Weinberg, 2019). We utilized the xenograft model with overexpressed BEST4 to analyze the lysates of tumor tissue, revealing that BEST4 upregulated E-cadherin and downregulated vimentin and Twist1 (Figures 2I). These findings indicate that BEST4 inhibits EMT in vivo. Deletion of BEST4 may enable these cells to acquire invasive and migratory abilities, leading to metastasis in vivo. Therefore, we subsequently evaluated the metastatic potential of BEST4 in a CRC liver metastasis model by intrasplenically injecting HCT-15 cell lines with knockout of BEST4 (BEST4gRNA), wild-type control (gRNA), or knockout with rescue (BEST4-Rescued) into BALB/c nude mice. Our observations revealed a twofold increase in liver metastatic nodules in the absence of BEST4 compared to the control group (Fig. 2J-L). Although further in vivo experiments are required for confirmation, our research suggests a potential role for BEST4 in counteracting EMT induction in vivo.

(7) In Figure 4H, ZO-1 and E-cad expression looks unchanged in the BEST4 KD.

Sincerely thanks for catching this issue. We have implemented the necessary modifications to the corresponding sections in the manuscript and performed a comprehensive quantification of all Western Blot data to ensure statistically significant differences, including those presented in the supplementary file.

(8) The in vivo and in vitro data supporting the whole transcriptome sequence analysis (Figure 5) is mostly insufficient. Including the following experiments will substantiate their claims: (1) BEST4 and HES4 immunostaining of human surgical tissue samples, (2) qPCR data of HES4, Twist1, Vimentin, etc. as shown in Figure 5C, 5D.

Sincerely thanks for catching this issue.

(1) Due to the substandard quality of the BEST4 antibody, we opted to evaluate the clinical significance of BEST4 in CRC by assessing mRNA results instead of protein levels using immunohistochemistry (IHC). After testing multiple antibodies for western blotting, only one (1:800; LsBio, LS-C31133) accurately indicated BEST4 protein expression while still exhibiting some non-specific bands. Consequently, we decided to transfect a HA-tagged BEST4 plasmid into the CRC cell line and used HA as a marker for BEST4 expression. Unfortunately, none of the antibodies employed for IHC were suitable as they failed to accurately distinguish between positive or negative staining for BEST4 and showed significant non-specific staining (data not shown). The challenge in detecting BEST4 protein in colorectal cancer tissues may be attributed to its low expression levels. Our findings are consistent with previous reports from the Human Protein Atlas database (https://www.proteinatlas.org/ENSG00000142959-BEST4/pathology), which also did not detect any BEST4 protein expression in colorectal cancer tissues through IHC analysis.

(2) The qPCR data of E-cadherin, Twist1, and Vimentin mRNA expression in CRC tissue has already been published in other studies(Christou et al., 2017; Lazarova and Bordonaro, 2016; Zhu *et al.*, 2015). It was found that E-cadherin is downregulated, while Twist1 and Vimentin are upregulated in CRC tissue compared to the adjacent normal tissues. The qPCR data of E-cadherin, Twist1, and Vimentin mRNA expression in CRC tissue has already been published in other studies(Christou *et al.*, 2017; Lazarova and Bordonaro, 2016; Zhu *et al.*, 2015). It was found that E-cadherin is downregulated, while Twist1 and Vimentin are upregulated in CRC tissue compared to the adjacent normal tissues. The analysis of mRNA expression data obtained from colorectal cancer samples and normal samples in the publicly available databases TCGA and GTEx also revealed a significant downregulation of _Hes_4 expression in colorectal cancer tissues, which will be our next research objective.

(9) Some statements are inconsistent probably due to grammatical errors. (For example, some High/low may be reversed in lines 234-244.)

We apologize for these mistakes. We have made corrections to this section in the manuscript.

**Reviewer #2 (Public Review):**
Summary:Using in vitro and in vivo approaches, the authors first demonstrate that BEST4 inhibits intestinal tumor cell growth and reduces their metastatic potential, possibly via downstream regulation of TWIST1.They further show that HES4 positively upregulates BEST4 expression, with direct interaction with BEST4 promoter region and protein. The authors further expand on this with results showing that negative regulation of TWIST1 by HES4 requires BEST4 protein, with BEST4 required for TWIST1 association with HES4. Reduction of BEST1 expression was shown in CRC tumor samples, with correlation of BEST4 mRNA levels with different clinicopathological factors such as sex, tumor stage, and lymph node metastasis, suggesting a tumor-suppressive role of BEST4 for intestinal cancer.Strengths:• Good quality western blot data.• Multiple approaches were used to validate the findings.• Logical experimental progression for readability.• Human patient data / In vivo murine model / Multiple cell lines were used, which supports translatability / reproducibility of findings.

We sincerely thank Reviewer #2 for constructive feedback on this work

Weaknesses:(1) Interpretation of figures and data (unsubstantiated conclusions).

We apologize for this confusing presentation. We have made corrections to this section in the manuscript.

(2) Figure quality.

We apologize for the poor quality of the figures. The figure resolution was drastically reduced during the conversion of the manuscript to pdf on publisher web site. The figures have been re-uploaded and we have once again confirmed that each image has a resolution exceeding 300dpi.

(3) Figure legends lack information.

Sincere thanks for catching this issue. We have provided detailed figure legends including supplementary figure legends on pages 36-43 of the manuscript. We have rechecked this section and made improvements and additions.

(4) Lacking/shallow discussion.

We apologize for our shallow discussion. We have supplemented and improved some parts of the discussion

(5) Requires more information for reproducibility regarding materials and methods.

Sincere thanks for catching this issue. We have provided detailed information for reproducibility regarding materials and methods on pages 18-29; 43-47 of the manuscript. We have rechecked this section and made improvements and additions.

**Recommendations for the authors:**

**Reviewer #1 (Recommendations For The Authors):**

We sincerely thank Reviewer #1 for constructive feedback on this work.

Minor comments:(1) Line 73: "Variant 4" is not precise. The term "variant" should mean mutation in the gene or different transcription.

We apologize for using an inaccurate expression. We have now changed Variant 4 to Bestrophin 4.

(2) Line 78. Is it correct that BEST4+ cells coexist with Hes4+ cells?

According to the previous study that published in Nature (Parikh et al., 2019), BEST4+ cells originate from the absorptive lineage and express the transcription factors Hes4. Additionally, we also observed the nuclear co-localization of BEST4 and Hes4 in HCT116 cells through immunofluorescence staining (Figure 3E)

(3) Line 85. The reason "Best4 may be associated with Twist1" is unclear.

We apologize for the lack of clarity in our previous statement. In a recent analysis utilizing single cell RNA-sequencing, it was discovered that a subset of mature colonocytes expresses BEST4 (Parikh et al., 2019). Additionally, this subset coexists with hairy/enhancer of split 4 positive (Hes4+) cells (Parikh et al., 2019). Previous research has demonstrated the antagonistic role of Hes4 in regulating Twist1 through protein-protein interaction, which governs the differentiation of bone marrow stromal/stem cell lineage (Cakouros et al., 2015). Based on these findings, we speculate that there may be an interactive regulation between BEST4/Hes4/Twist1, potentially influencing the process of cell polarity during epithelial-mesenchymal transition in colorectal cancer. We have made corrections to this section in the manuscript.

(4) Line 87. Grammatical error (Establishing the role BEST4).

We apologize for the grammatical error of this section. We have rectified the issue in the manuscript.

(5) Please clarify the reason the authors do not show any data of BEST4-overexpressing Caco2 cells in Figure 2?

We apologize for our negligence in not adding this data to in Figure 2. It has now been fully supplemented.

(6) In line 145, the authors did not show any tumorigenic properties.

We apologize for this confusing presentation. We have made corrections to this section in the manuscript.

(7) Figure 3 shows (1) HES4 regulates BEST4 promotor activity, and (2) HES4 and BEST4 colocalized in nuclei, but these are very different biological processes. Please clarify how these two relate to each other.

Trajectory analysis identifies the basic helix-loop-helix (bHLH) transcription factors Hairy/enhancer of split 4 (Hes4)-expressing colonocytes (Hes4+) in BEST4-expressing colonic epithelial lineage (BEST4+). Although they are very different biological processes, the recent identification of a heterogeneous *BEST4*+ and *Hes4*+ subgroup in a human colonic epithelial lineage (Parikh *et al.*, 2019) led us to consider their potential role in regulating CRC progression. We firstly observed a responsive upregulation of both endogenous *BEST4* mRNA and protein levels in Hes4 overexpression cells compared to the control transfectant, indicating that Hes4 is a potential upstream activator regulating BEST4 functional. We then confirmed that Hes4 interacted with BEST4, binding directly to its upstream promoter at the region of 1470-1569 bp enhancing its promoter activity as analysed by Co-IP, dual-luciferase assay and ChIP-qPCR, respectively. Essentially, they were co-localized in the nucleus, as shown in immunofluorescence staining after the transient co-transfection of Hes4 and BEST4 into HCT116, therefore indicating that *BEST4* interacts with *Hes4* at both transcriptional and translational levels (Figure 3; Figure 3-supplemental figure 1)

(8) In line 182-185, please clarify the reason BEST4 mediates the inhibition of the Twist 1 promotor activity by Hes4.

Because a step of *Hes4* in committing to human bone marrow stromal/stem cell lineage-specific development is mediated by Twist1 downregulation (Cakouros et al., 2015), with evidence of direct interaction between BEST4 and Hes4 observed in HCT116, it is plausible that they could exploit Twist1 to regulate EMT. In the present study, we found that Twist1 colocalized with BEST4 in the nucleus, and their interaction destabilized Twist1 and significantly inhibited EMT induction. *Hes4* caused the same effect; however, it required intermediation through *BEST4*. Although the mechanistic insights of their intercorrelation remain to be elucidated, the present study identified the axis of Hes4-BEST4-Twist1 governing the development of CRC, at least partially by counteracting EMT induction

(9) In line 205, please rephrase "BEST4-mediated upstream Hes4" to be clearer.

We apologize for this confusing presentation. We have made corrections to this section in the manuscript.

**Reviewer #2 (Recommendations For The Authors):**

We sincerely thank Reviewer #2 for constructive feedback on this work

Major Comments:(1) The general quality of the figures requires improvement (text in some figures is illegible, and the resolution of the images is low) with more proofreading of the text for clarity. In addition, the resolution of the histology in Fig 2K does not allow a proper evalution of the data.

We apologize for the poor quality of the figures. The figure resolution was drastically reduced during the conversion of the manuscript to pdf on publisher web site. The figures have been re-uploaded and we have once again confirmed that each image has a resolution exceeding 300dpi. Meanwhile, the Figure 2K was further enhanced and expanded.

(2) While the authors show that the HES4/BEST4 complex interacts with the TWIST1 protein, they do not expand on the mechanisms underpinning the post-translational or transcriptional regulation of TWIST1. We would like the authors to prove or further speculate on the mechanisms behind this regulation in the discussion.

Our present study showed that BEST4 inhibited EMT in conjunction with downregulation of Twist1 in both HCT116 and Caco2 CRC cell lines. A previous study has shown an antagonist role of Hes4 in regulating Twist1 via protein-protein interaction that controls the bone marrow stromal/stem cell lineage differentiation (Cakouros *et al.*, 2015). We speculate a possible interactive regulation between Hes4/BEST4/Twist1 by which they deter the process of cell polarity during EMT in CRC. In the present study, we found that *BEST4* mediates the inhibition of the Twist1 both in transcription and translation level by *Hes4.* Twist1 colocalized with BEST4 in the nucleus, and their interaction destabilized Twist1 and significantly inhibited EMT induction. *Hes4* caused the same effect; however, it required intermediation through *BEST4*. The present study identified the axis of Hes4-BEST4-Twist1 governing the development of CRC, at least partially by counteracting EMT induction. We agree that further studies to elucidate mechanistic insights of their intercorrelation are needed that are beyond the scope of the current work.

(3) The authors need to show or argue that why TWIST1 is necessary for the phenotypes observed, e.g. metastasis/proliferation.

We apologize for the lack of clarity in articulating this question. The process of EMT transforms epithelial cells exhibiting a spindle fibroblast-like morphology, leading to the acquisition of mesenchymal characteristics and morphology, enabling these cells to acquire invasive and migratory abilities, with expression switching epithelial E-cadherin and Zo-1 to mesenchymal vimentin (Dongre and Weinberg, 2019). When diagnosed in advanced stages, EMT may occur as CRC metastasize to distal organs (Pastushenko and Blanpain, 2019; Sunlin Yong, 2021; Yeung and Yang, 2017; Zhang et al., 2021).The whole process is regulated by transcriptional factors of the Snail family and Twist1(Dongre and Weinberg, 2019). Twist1 (a basic helix-loop-helix transcription factor) reprograms EMT by repressing the expression of E-cadherin and ZO-1 (Nagai et al., 2016; Yang et al., 2004) and simultaneously inducing several mesenchymal markers, typically vimentin (Bulzico et al., 2019; Meng et al., 2018; Nagai *et al.*, 2016; Yang *et al.*, 2004), which is a pivotal predictor of CRC progression (Vesuna et al., 2008; Yang *et al.*, 2004; Yusup et al., 2017; Zhu et al., 2015).Overexpression of Twist1 significantly enhances the migration and invasion capabilities of colorectal cancer cells; furthermore, it is closely associated with metastasis and poor prognosis in patients with colorectal cancer(Yusup *et al.*, 2017; Zhu *et al.*, 2015). We have supplemented and improved these parts of the introduction and discussion.

(4) The authors sufficiently prove that HES4/BEST4 regulates TWIST1 downregulation, however, we believe the findings are not enough to show *direct* regulation (refer also to line 273). At least rephrasing the conclusions would be adequate, also while referring to the working model depicted in Fig. 5G.

We apologize for this inaccurate presentation. Although the interaction may not be direct, our co-immunoprecipitation (CO-IP) results demonstrated nuclear colocalization of Twist1 and BEST4 (Figure 4D; Figure 4-supplemental figure 1A). Furthermore, their interaction destabilized Twist1 and significantly inhibited the induction of EMT. We have made corrections to this section in the manuscript.

(5) The discussion is very short and not satisfactory; is BEST4 an evolutionary conserved protein (besides the channel region)? Any speculation on which domain(s) is(are) important for the interaction with HES4 and TWIST1? How do the findings in the current study compare with recent, potentially contradicting data indicating a pro-tumorigenic function of BEST4 for CRC, including its upregulation (and not downregulation) in malignant intestinal tissues, and activation of PI3K/AKT signaling (PMID: 35058597)?

We apologize for our shallow discussion. We have supplemented and improved some parts of the discussion. The bestrophins are a highly conserved family of integral membrane proteins initially discovered in *Caenorhabditis elegans* (Sonnhammer and Durbin, 1997). Homologous sequences can be found across animals, fungi, and prokaryotes, while they are absent in protozoans or plants(Hagen et al., 2005). Conservation is primarily observed within the N-terminal 350–400 amino acids, featuring an invariant motif arginine-phenylalanine-proline (RFP) with unknown functional properties (Milenkovic et al., 2008). Mutations in this region can lead to the development of vitelliform macular dystrophy. However, the C-terminus is a potential site for protein modification and function(Marmorstein et al., 2002; Miller et al., 2019). There is currently no further literature research on the functional roles of different domains of BEST4. Although the crucial domain for the interaction with HES4 and TWIST1 is yet to be determined, requiring further investigation for clarification, our findings demonstrate that Hes4 directly binds to the upstream promoter region of BEST4 at 1470-1569 bp, thereby enhancing its promoter activity. These results provide valuable insights for future research.

Sincere thanks for catching this publication to us. We carefully read this study and would like to point out a few things.

a) Firstly, the study demonstrated that BEST4 expression is upregulated in clinical CRC samples, which contradicts the results of other published studies except for our own research. RNA-seq of tissue samples from 95 human individuals representing 27 different tissues was performed to determine the tissue specificity of all protein-coding genes, and the results indicated that the BEST4 gene is predominantly expressed in the colon (Fagerberg et al., 2014). In addition, BEST4 was reported to be exclusively expressed by human absorptive cells and could be induced during the process of human absorptive cell differentiation(Ito et al., 2013). Recently, the research from Simmons’s group that published in Nature further proved that human absorptive colonocytes distinctly express BEST4 by single-cell profiling of healthy human colonic epithelial cells, and is dysregulated in colorectal cancer patients(Parikh *et al.*, 2019). Furthermore, the analysis of RNA-seq expression data obtained from colorectal cancer samples and normal samples in the publicly available databases TCGA and GTEx also revealed a significant downregulation of BEST4 expression in colorectal cancer tissues, which is consistent with our research findings. The literature above demonstrates a close relationship between BEST4 and the normal function of the human colon, and provide evidence for their loss in colorectal cancer patients.

b) Their study showed an increased expression of BEST4 protein levels in colorectal cancer patients through Western Blot. However, the antibody they used was only suitable for IHC-P and not for Western Blot (Abcam , ab188823); . In our study, we also utilized WB technology to detect the expression of BEST4 in colorectal cancer tissues and adjacent normal tissues. The results revealed a decreased expression of BEST4 protein levels in colorectal cancer patients. The antibody we used was specifically designed for WB detection (1:800; LsBio, LS-C31133 https://www.lsbio.com/antibodies/best4-antibody-n-terminus-wb-western-ls-c31133/29602).

c) The study demonstrated an upregulation of BEST4 protein levels in colorectal cancer patients using immunohistochemistry (IHC). However, the expression of BEST4 was assessed in colorectal cancer tissues through IHC utilizing publicly available protein expression databases such as the Human Protein Atlas. Interestingly, this analysis revealed a minimal presence of BEST4 protein in colorectal cancer tissues (https://www.proteinatlas.org/ENSG00000142959-BEST4/pathology), contradicting their research findings but aligning with our own observations.

d) Literature based on single-cell genomics analysis reports that only OTOP2 and BEST4 genes are expressed in a subset of the normal colorectal epithelial cells but not the rest(Parikh *et al.*, 2019). An inhibitory effect of OTOP2 on CRC has been recently shown BEST4, and the Otopetrin 2 (OTOP2), which encodes proton‐selective ion channel protein were reported to distinct expressed in normal absorptive colonocytes and colocalized with each other (Drummond et al., 2017; Ito *et al.*, 2013; Parikh *et al.*, 2019). OTOP2 has been recently demonstrated to have an inhibitory effect on the development of CRC via being regulated by wide-type p53(Qu et al., 2019), while the role of BEST4 in CRC is less well studied, that indicate the potential of BEST4 to inhibit colorectal cancer. The Gene set enrichment analysis (GSEA) conducted by them revealed a significant enrichment of gene signatures associated with oncogenic signaling and metastasis, such as the PI3K/Akt signaling pathway, in patients exhibiting higher BEST4 expression compared to those with lower BEST4 expression. However, our GSEA did not show any significant enrichment of the PI3K/Akt signaling pathway in patients with higher BEST4 expression compared to those with lower BEST4 expression. In contrast to their findings, our BEST4 overexpression cell line did not exhibit a significant increase in phosphorylated Akt levels. The present study concludes that our findings align with previous literature and public database analyses, providing evidence for the downregulation of BEST4 in colorectal cancer tissues and its potential as an anticancer agent. Discrepancies observed in other studies may be attributed to difference in experimental model, protocols, preparations or experimental conditions.

Minor Comments:(1) Western blot data should be quantified.

Sincere thanks for catching this point to us. We have conducted a comprehensive quantification of all the Western Blot data and included the results in the supplementary file.

(2) Errors in labelling figures in the text should be corrected (Line 214 and more).

We apologize for these mistakes. We have made corrections to this section in the manuscript.

(3) The authors used the human HES4 gene, which is indicated with the incorrect nomenclature. The gene and protein nomenclature should be correctly used.

We apologize for these mistakes. We have made corrections to this section in the manuscript.

(4) Methods and Materials for certain assays should be further clarified; e.g transwell migration/invasion assays (reference to previous publication? transwell inserts used, etc.)

Sincerely thanks for catching this issue. We have implemented enhancements and updates to the respective sections.

(5) Figure 2K: Quality of histology is insufficient.

We apologize for the poor quality of the figures. The quality of Figure 2K was further enhanced and expanded.

(6) Figure 2K: Can the authors speculate on whether there is any increase in proliferation through BEST4-ko in HCT15 cells (with overexpression of BEST4 leading to reduced proliferation) and how this may impact the metastatic assay or engraftment/seeding onto the liver?

Our in vitro experiment demonstrated that the ablation of BEST4 in HCT-15 cells resulted in increased cell proliferation, clonogenesis, migration and invasion (Figures 1 and Figure 1-supplemental figure 1). These findings suggest that BEST4 knockout may potentially contribute to tumor proliferation in vivo; however, further research is required for confirmation. EMT transforms epithelial cells exhibiting a spindle fibroblast-like morphology, leading to the acquisition of mesenchymal characteristics and morphology, enabling these cells to acquire invasive and migratory abilities (Dongre and Weinberg, 2019). When diagnosed in advanced stages, EMT may occur as CRC metastasize to distal organs (Pastushenko and Blanpain, 2019; Sunlin Yong, 2021; Yeung and Yang, 2017; Zhang *et al.*, 2021). Our study demonstrated that BEST4 inhibits EMT in colorectal cancer (CRC) both in vitro and in vivo. Conversely, ablation of BEST4 promotes EMT by upregulating the expression of EMT-related genes, thereby facilitating the metastasis of colorectal cancer cells to the liver.

(7) Figure 2L: Authors should indicate in the figure that the BEST4-rescued is at 0 (and not blank).

Sincerely thanks for catching this issue. We have made corrections to this section in the manuscript.

(8) Figure 3B: Authors should introduce the usage of the new LS174T cell line in the text.

Sincerely thanks for catching this issue. The human colorectal cancer cell line, LS174T, was selected for Hes4 knockdown due to its comparatively higher expression of Hes4 in comparison to other CRC cell lines. We have made corrections to this section in the manuscript.

(9) Figure 3F: Why is there less FLAG in the input, compared to the IP?

Sincerely thanks for catching this issue. Cell lysates (20 µg) were used for input, and 500ug for IP according to the manufacturer's protocols.

(10) Figure 5F-G: the quality of the figure is not good enough for interpretation.

Again, we apologize for poor quality of pictures due to manuscript conversion. We have made corrections to this section in the manuscript.

(11) Table 1: Conclusions made by the authors are wrong (lines 237-239); instead "high BEST4 expression more prevalent in females" and "low BEST4 expression more prevalent among CRC patients with advanced tumor stage". And how are low and high BEST4 expressions defined (the same applies to the data in Fig. 5F)?

We apologize for these mistakes, we set cutoff-*high* (*50*%) and cutoff-*low* (*50*%) values to split the *high*-*expression* and *low-expression* cohorts. We have made corrections to this section in the manuscript.

(12) In all Figure legends, there should be an indication of the type of statistical tests that were applied, as well as information on the number of independent experiments that were performed and provided the same results

Sincerely thanks for catching this issue. The types of statistical tests applied in the Materials and Method- Statistical analysis section are indicated. Information on the number of independent experiments used is provided in the figure legend section.

Reference

Bulzico, D., Pires, B.R.B., PAS, D.E.F., Neto, L.V., and Abdelhay, E. (2019). "Twist1 Correlates With Epithelial-Mesenchymal Transition Markers Fibronectin and Vimentin in Adrenocortical Tumors". Anticancer research *39*, 173-175. 10.21873/anticanres.13094.

Cakouros, D., Isenmann, S., Hemming, S.E., Menicanin, D., Camp, E., Zannetinno, A.C., and Gronthos, S. (2015). "Novel basic helix-loop-helix transcription factor hes4 antagonizes the function of twist-1 to regulate lineage commitment of bone marrow stromal/stem cells". Stem Cells Dev *24*, 1297-1308. 10.1089/scd.2014.0471.

Christou, N., Perraud, A., Blondy, S., Jauberteau, M.O., Battu, S., and Mathonnet, M. (2017). "E-cadherin: A potential biomarker of colorectal cancer prognosis". Oncol Lett *13*, 4571-4576. 10.3892/ol.2017.6063.

Dongre, A., and Weinberg, R.A. (2019). "New insights into the mechanisms of epithelial-mesenchymal transition and implications for cancer". Nature reviews. Molecular cell biology *20*, 69-84. 10.1038/s41580-018-0080-4.

Drummond, C.G., Bolock, A.M., Ma, C., Luke, C.J., Good, M., and Coyne, C.B. (2017). "Enteroviruses infect human enteroids and induce antiviral signaling in a cell lineage-specific manner". Proceedings of the National Academy of Sciences of the United States of America *114*, 1672-1677. 10.1073/pnas.1617363114.

Fagerberg, L., Hallstrom, B.M., Oksvold, P., Kampf, C., Djureinovic, D., Odeberg, J., Habuka, M., Tahmasebpoor, S., Danielsson, A., Edlund, K., et al. (2014). "Analysis of the human tissue-specific expression by genome-wide integration of transcriptomics and antibody-based proteomics". Mol Cell Proteomics *13*, 397-406. 10.1074/mcp.M113.035600.

Hagen, A.R., Barabote, R.D., and Saier, M.H. (2005). "The bestrophin family of anion channels: identification of prokaryotic homologues". Molecular membrane biology *22*, 291-302. 10.1080/09687860500129711.

Ito, G., Okamoto, R., Murano, T., Shimizu, H., Fujii, S., Nakata, T., Mizutani, T., Yui, S., Akiyama-Morio, J., Nemoto, Y., et al. (2013). "Lineage-specific expression of bestrophin-2 and bestrophin-4 in human intestinal epithelial cells". PLoS One *8*, e79693. 10.1371/journal.pone.0079693.

Lazarova, D.L., and Bordonaro, M. (2016). "Vimentin, colon cancer progression and resistance to butyrate and other HDACis". Journal of cellular and molecular medicine *20*, 989-993. 10.1111/jcmm.12850.

Marmorstein, L.Y., McLaughlin, P.J., Stanton, J.B., Yan, L., Crabb, J.W., and Marmorstein, A.D. (2002). "Bestrophin interacts physically and functionally with protein phosphatase 2A". The Journal of biological chemistry *277*, 30591-30597. 10.1074/jbc.M204269200.

Meng, J., Chen, S., Han, J.X., Qian, B., Wang, X.R., Zhong, W.L., Qin, Y., Zhang, H., Gao, W.F., Lei, Y.Y., et al. (2018). "Twist1 Regulates Vimentin through Cul2 Circular RNA to Promote EMT in Hepatocellular Carcinoma". Cancer research *78*, 4150-4162. 10.1158/0008-5472.Can-17-3009.

Milenkovic, V.M., Langmann, T., Schreiber, R., Kunzelmann, K., and Weber, B.H. (2008). "Molecular evolution and functional divergence of the bestrophin protein family". BMC evolutionary biology *8*, 72. 10.1186/1471-2148-8-72.

Miller, A.N., Vaisey, G., and Long, S.B. (2019). "Molecular mechanisms of gating in the calcium-activated chloride channel bestrophin". eLife *8*. 10.7554/eLife.43231.

Nagai, T., Arao, T., Nishio, K., Matsumoto, K., Hagiwara, S., Sakurai, T., Minami, Y., Ida, H., Ueshima, K., Nishida, N., et al. (2016). "Impact of Tight Junction Protein ZO-1 and TWIST Expression on Postoperative Survival of Patients with Hepatocellular Carcinoma". Digestive diseases (Basel, Switzerland) *34*, 702-707. 10.1159/000448860.

Parikh, K., Antanaviciute, A., Fawkner-Corbett, D., Jagielowicz, M., Aulicino, A., Lagerholm, C., Davis, S., Kinchen, J., Chen, H.H., Alham, N.K., et al. (2019). "Colonic epithelial cell diversity in health and inflammatory bowel disease". Nature *567*, 49-55. 10.1038/s41586-019-0992-y.

Pastushenko, I., and Blanpain, C. (2019). "EMT Transition States during Tumor Progression and Metastasis". Trends in cell biology *29*, 212-226. 10.1016/j.tcb.2018.12.001.

Qu, H., Su, Y., Yu, L., Zhao, H., and Xin, C. (2019). "Wild-type p53 regulates OTOP2 transcription through DNA loop alteration of the promoter in colorectal cancer". FEBS open bio *9*, 26-34. 10.1002/2211-5463.12554.

Sonnhammer, E.L., and Durbin, R. (1997). "Analysis of protein domain families in *Caenorhabditis elegans*". Genomics *46*, 200-216. 10.1006/geno.1997.4989.

Sunlin Yong, Z.W., Tang Yuan, Chuang Cheng, Dan Jiang (2021). "Comparison of MMR protein and Microsatellite Instability Detection in Colorectal Cancer and Its Clinicopathological Features Analysis". Journal of Medical Research *50*, 61-66. 10.11969/j.issn.1673-548X.2021.05.015

Vesuna, F., van Diest, P., Chen, J.H., and Raman, V. (2008). "Twist is a transcriptional repressor of E-cadherin gene expression in breast cancer". Biochem Biophys Res Commun *367*, 235-241. 10.1016/j.bbrc.2007.11.151.

Yang, J., Mani, S.A., Donaher, J.L., Ramaswamy, S., Itzykson, R.A., Come, C., Savagner, P., Gitelman, I., Richardson, A., and Weinberg, R.A. (2004). "Twist, a master regulator of morphogenesis, plays an essential role in tumor metastasis". Cell *117*, 927-939. 10.1016/j.cell.2004.06.006.

Yeung, K.T., and Yang, J. (2017). "Epithelial-mesenchymal transition in tumor metastasis". Molecular oncology *11*, 28-39. 10.1002/1878-0261.12017.

Yusup, A., Huji, B., Fang, C., Wang, F., Dadihan, T., Wang, H.J., and Upur, H. (2017). "Expression of trefoil factors and TWIST1 in colorectal cancer and their correlation with metastatic potential and prognosis". World journal of gastroenterology *23*, 110-120. 10.3748/wjg.v23.i1.110.

Zhang, N., Ng, A.S., Cai, S., Li, Q., Yang, L., and Kerr, D. (2021). "Novel therapeutic strategies: targeting epithelial-mesenchymal transition in colorectal cancer". The Lancet. Oncology *22*, e358-e368. 10.1016/s1470-2045(21)00343-0.

Zhu, D.J., Chen, X.W., Zhang, W.J., Wang, J.Z., Ouyang, M.Z., Zhong, Q., and Liu, C.C. (2015). "Twist1 is a potential prognostic marker for colorectal cancer and associated with chemoresistance". American journal of cancer research *5*, 2000-2011.